# Post-Hoc Reversal: Are We Selecting Models Prematurely?

**Rishabh Ranjan**[1]*, **Saurabh Garg**[2], **Mrigank Raman**[2], **Carlos Guestrin**[1,3], **Zachary Lipton**[2]

[1]Stanford University, [2]Carnegie Mellon University, [3]Chan Zuckerberg Biohub

{ranjanr,guestrin}@stanford.edu, {sgarg2,mrigankr,zlipton}@cmu.edu

## Abstract

Trained models are often composed with post-hoc transforms such as temperature scaling (TS), ensembling and stochastic weight averaging (SWA) to improve performance, robustness, uncertainty estimation, etc. However, such transforms are typically applied only after the base models have already been finalized by standard means. In this paper, we challenge this practice with an extensive empirical study. In particular, we demonstrate a phenomenon that we call *post-hoc reversal*, where performance trends are reversed after applying post-hoc transforms. This phenomenon is especially prominent in high-noise settings. For example, while base models overfit badly early in training, both ensembling and SWA favor base models trained for more epochs. Post-hoc reversal can also prevent the appearance of double descent and mitigate mismatches between test loss and test error seen in base models. Preliminary analyses suggest that these transforms induce reversal by suppressing the influence of mislabeled examples, exploiting differences in their learning dynamics from those of clean examples. Based on our findings, we propose *post-hoc selection*, a simple technique whereby post-hoc metrics inform model development decisions such as early stopping, checkpointing, and broader hyperparameter choices. Our experiments span real-world vision, language, tabular and graph datasets. On an LLM instruction tuning dataset, post-hoc selection results in $> 1.5\times$ MMLU improvement compared to naive selection.[2]

## 1 Introduction

Many widely used techniques in deep learning operate on trained models; we refer to these as *post-hoc transforms*. Examples include temperature scaling (TS) [19], stochastic weight averaging (SWA) [28] and ensembling [39]. These techniques have shown promise for improving predictive performance, robustness, uncertainty estimation, out-of-distribution generalization, and few-shot performance [4, 6, 39, 56, 84]. Typically, the pre-training and post-hoc stages are isolated. The workflow is: (1) pick model architecture, training recipe, hyperparameters, etc. to optimize for individual model performance; (2) train one or more models; (3) pick best-performing checkpoints; (4) apply post-hoc transforms. We refer to this procedure as *naive selection*.

In this paper, we demonstrate interesting drawbacks of naive selection. In a large-scale empirical study, we uncover *post-hoc reversal*—a phenomenon whereby post-hoc transforms reverse performance trends between models (Fig. 1). We demonstrate post-hoc reversal with respect to training epochs, model sizes, and other hyperparameters like learning rate schedules. We further establish that post-hoc reversal is a robust phenomenon by experimenting on real-world datasets across domains and modalities, with diverse model classes and training setups.

38th Conference on Neural Information Processing Systems (NeurIPS 2024).

---

*Undertaken in part as a visiting researcher at Carnegie Mellon University.

[2]Code is available at `https://github.com/rishabh-ranjan/post-hoc-reversal`.

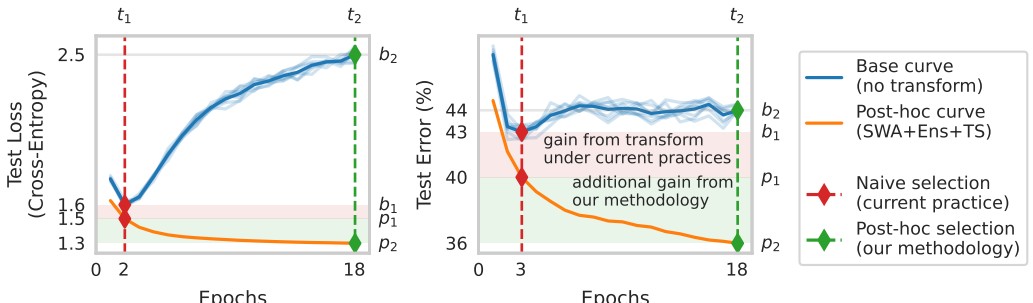

Figure 1: An illustration of the *phenomenon* of *post-hoc reversal* on the FMoW dataset: base performance at epoch $t_2$ is worse than at epoch $t_1$ ($b_2 > b_1$), but post-hoc performance is better ($p_2 < p_1$). The current practice of *naive selection* considers base metrics to pick models at epoch $t_1$. Our proposed *technique* of *post-hoc selection* instead uses post-hoc metrics to pick models at epoch $t_2$, resulting in $> 2\times$ improvement over naive selection in both test loss and error. SWA+Ens+TS refers to the post-hoc transform obtained by composing SWA, ensemble (Ens) and temperature scaling (TS). Base curves show mean of $8$ runs, models from which constitute the ensembles. Individual runs are shown in lighter colors. See Fig. 5 for more detailed curves on this dataset.

Post-hoc reversal is most prominent on noisy datasets (Fig. 2). Other phenomena exacerbated by noise include catastrophic overfitting [50],double descent [55], and loss-error mismatch [19]. While these phenomena pose challenges to model development, post-hoc reversal suggests a path to alleviate them. Noise can arise not only from labeling errors, but also from inherent uncertainty in the prediction task, such as in next token prediction [60]. Indeed, severe performance degradation has limited multi-epoch training of large language models (LLMs) [81]. Here too, post-hoc reversal reveals a promising path for sustained performance improvements over longer training.

The core intuition for post-hoc reversal is that models continue to learn generalizable patterns from clean examples, even when spurious patterns learnt from mislabeled examples worsen the overall performance. Post-hoc transforms exploit differences in the learning dynamics of clean and mislabeled examples [42] to reinforce the influence of the former, while suppressing that of the latter. When strong enough, this effect leads to reversal. We show evidence for these intuitions in § 5.

Based on our findings, we propose *post-hoc selection*—a simple technique whereby base models are selected based on post-transform performance. The technique is practical as the transforms of interest can be cheaply incorporated into the validation phase of the training loop. Post-hoc selection significantly improves the performance of the transformed models, with $> 2\times$ improvements over naive selection in some cases (Fig. 2). In terms of absolute performance, post-hoc selection leads to $> 3$-point reduction in test error over naive selection on a satellite imaging dataset (Fig. 1). The reduction is even higher ($> 5$ points) when using out-of-distribution (OOD) val/test splits for the same dataset. On an LLM instruction tuning dataset, under our procedure a composed transform of SWA, ensemble and TS gives $> 1.5\times$ MMLU improvement over a naive application of the same transform on prematurely selected models.

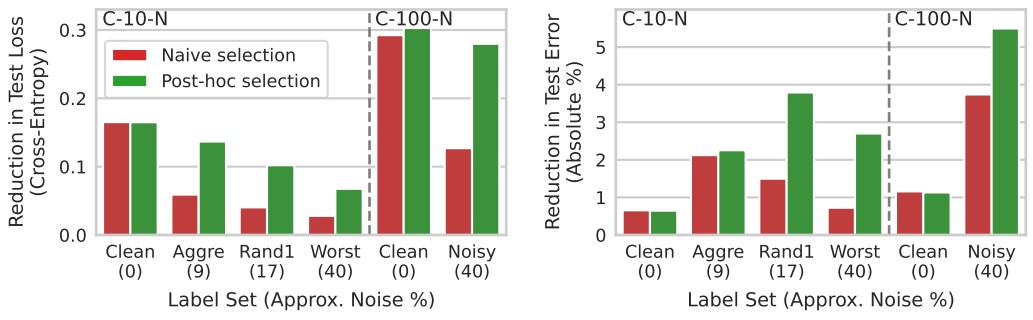

Figure 2: A comparison of naive and post-hoc selection on label sets from CIFAR-10/100-N (abbr. C-10/100-N) for the SWA+TS transform. On noisy label sets, post-hoc selection is often $> 2\times$ better.

## 2 Related Work

A slew of empirical works [10, 17, 31, 55, 57, 58] have revealed both challenges and opportunities for improving the understanding and practice of deep learning. Our work expands this list with a novel phenomenon tying together noisy data learning and post-hoc transforms. Orthogonal to our work, a number of training-stage strategies for noisy data have been proposed (see [69] for a survey).

TS belongs to a family of calibration techniques [2, 19] proposed with the goal of producing well-calibrated probabilities. Ensembling is a foundational technique in machine learning, with simple variants routinely used in deep learning [3, 39]. SWA [28] is the culmination of a line of work [18, 25] seeking to cheaply approximate ensembling. Despite their prevalence, a thorough understanding of best practices for wielding these techniques is lacking, especially in the context of noisy data. Our work fills this gap. For a more detailed discussion on related work, see App. A.

## 3 Preliminaries and Background

We describe our learning setup in § 3.1, with emphasis on noisy data, a key focus of this work. In § 3.2, we introduce the post-hoc transforms we study.

### 3.1 Learning on Noisy Data

**Setup.** We consider multi-class classification with $C$ classes, input $\mathbf{x} \in \mathcal{X}$ and label $y \in \mathcal{Y} = \{1, \ldots, C\}$. Training, validation and test sets are drawn i.i.d. from the data distribution $\mathcal{D}$. A *classifier* $f \colon \Theta \times \mathcal{X} \to \mathbb{R}^C$ outputs the logit vector $\mathbf{z} = f(\mathbf{x}; \boldsymbol{\theta})$, given parameter vector $\boldsymbol{\theta} \in \Theta$. Predicted probability of class $k$ is $\mathbb{P}_f[y = k \mid \mathbf{x}] = \sigma(\mathbf{z})_k$, where $\sigma$ is the softmax function.

**Noise.** Data $\mathcal{D}$ is said to be *clean* if $\mathbb{P}_{\mathcal{D}}[y \mid \mathbf{x}]$ is one-hot for all $\mathbf{x}$, *i.e.*, $\mathbb{P}_{\mathcal{D}}[y \mid \mathbf{x}] = \mathbf{1}\{y = y^*(\mathbf{x})\}$ for some labeling function $y^* \colon \mathcal{X} \to \mathcal{Y}$. Then, for any example input $\mathbf{x}^{(i)}$ in the dataset, the observed label is $y^{(i)} = y^*(\mathbf{x}^{(i)})$. When $\mathbb{P}_{\mathcal{D}}[y \mid \mathbf{x}]$ is not one-hot, $\mathcal{D}$ is said to be *noisy* and the observed label is only a stochastic sample $y^{(i)} \sim \mathbb{P}_{\mathcal{D}}[y \mid \mathbf{x} = \mathbf{x}^{(i)}]$ from the underlying conditional distribution. Noise can arise due to (1) non-determinism in the prediction target (2) insufficient information in the input context, and (3) annotation errors. See App. B.1 for illustrated examples.

**Metrics.** A metric $\mathcal{M} \colon \mathbb{R}^C \times \mathcal{Y} \to \mathbb{R}$ compares the predicted logits $\mathbf{z}$ with the observed label $y$. $\mathcal{M}_f(\boldsymbol{\theta}) = \mathcal{M}[f(\,\cdot\,; \boldsymbol{\theta})] = \mathbb{E}_{(\mathbf{x},y)\sim\mathcal{D}}[\mathcal{M}(f(\mathbf{x}; \boldsymbol{\theta}), y)]$ denotes the metric computed over $\mathcal{D}$ given $f$ and $\boldsymbol{\theta}$. We use two metrics (1) *classification error*, or simply *error*, with $\mathcal{M}^{\text{error}}(\mathbf{z}, y) = \mathbf{1}\{\arg\max_k \mathbf{z}_k \neq y\}$ and (2) *cross-entropy loss*, or simply *loss*, with $\mathcal{M}^{\text{loss}}(\mathbf{z}, y) = -\log \sigma(\mathbf{z})_y$. The exponentiated loss, also called *perplexity*, is common in language modeling, where it is computed on a per-token basis. A standard result states that loss is minimized if and only if the ground truth conditional probability is recovered [20]. See App. B.1 for additional background.

### 3.2 Post-Hoc Transforms in Machine Learning

**Definition 1 (Post-Hoc Transform)** *A post-hoc transform $\mathcal{T}$ maps a classifier $f \colon \Theta \times \mathcal{X} \to \mathcal{Y}$ to another classifier $\mathcal{T} \circ f \colon \Theta^K \times \mathcal{X} \to \mathcal{Y}$, for some $K$.*

**Temperature Scaling (TS).** TS [19] involves scaling the logits with a *temperature* $\tau \in \mathbb{R}$ obtained by optimizing the cross-entropy loss over the validation set, with model parameters fixed (Eqn. 1). Temperature scaling preserves error as it does not affect the predicted class. We use the `torchcal` [63] implementation, which optimizes the temperature on GPU with Newton's method [15].

$$(\mathcal{T}_{\text{TS}} \circ f)(\mathbf{x}; \boldsymbol{\theta}) = \frac{1}{\tau} f(\mathbf{x}; \boldsymbol{\theta}), \text{ with } \tau = \arg\min_\tau \mathcal{M}^{\text{loss}}_{\text{val}} \left[ \frac{1}{\tau} f(\,\cdot\,; \boldsymbol{\theta}) \right] \tag{1}$$

**Ensembling.** In this method, predictions from an ensemble of classifiers are combined. In deep learning, simply averaging the temperature-scaled logits is effective (Eqn. 2). $\boldsymbol{\theta}_1, \ldots, \boldsymbol{\theta}_K$ are obtained from multiple training runs with the same architecture and dataset, with stochasticity from mini-batch sampling and random initialization, if applicable.

$$(\mathcal{T}_{\text{Ens}} \circ f)(\mathbf{x}; \boldsymbol{\theta}_1, \ldots, \boldsymbol{\theta}_K) = \frac{1}{K} \sum_{k=1}^{K} \frac{1}{\tau_k} f(\mathbf{x}; \boldsymbol{\theta}_k), \text{ with } \tau_k = \arg\min_{\tau} \mathcal{M}_{\text{val}}^{\text{loss}} \left[ \frac{1}{\tau} f(\,\cdot\,; \boldsymbol{\theta}_k) \right] \quad (2)$$

**Stochastic Weight Averaging (SWA).** SWA [28] involves averaging weights $\boldsymbol{\theta}_1, \ldots, \boldsymbol{\theta}_K$ from the same training run (Eqn. 3). BatchNorm statistics are recomputed after averaging, if required. We pick checkpoints at epoch boundaries. Unlike Izmailov et al. [28], we do not skip the initial epochs (warmup) or modify the learning rate schedule[3].

$$(\mathcal{T}_{\text{SWA}} \circ f)(\mathbf{x}; \boldsymbol{\theta}_1, \ldots, \boldsymbol{\theta}_K) = f\left( \mathbf{x}; \frac{1}{K} \sum_{i=1}^{K} \boldsymbol{\theta}_i \right) \quad (3)$$

**Compositions.** TS, ensembling and SWA can be readily composed. In particular, we consider *SWA+TS* and *SWA+Ens+TS*, for single- and multi-model settings respectively. We denote them with $\mathcal{T}_{\text{S+T}} = \mathcal{T}_{\text{TS}} \circ \mathcal{T}_{\text{SWA}}$ and $\mathcal{T}_{\text{S+E+T}} = \mathcal{T}_{\text{TS}} \circ \mathcal{T}_{\text{Ens}} \circ \mathcal{T}_{\text{SWA}}$ (explicit forms in App. B.2).

# 4 Post-Hoc Reversal: Formalization and Empirical Study

To use post-hoc transforms, one must first select models to apply them to. Current practice is to select the best-performing model independent of post-hoc transforms, rationalized by an implicit *monotonicity* assumption – "better-performing models result in better performance after transformation". As we shall see, this assumption is often violated in practice. We call such violations *post-hoc reversal*. In § 4.1, we formalize post-hoc reversal and discuss ways to detect it. In § 4.2, we empirically study various kinds of post-hoc reversal with special practical relevance.

## 4.1 Definitions

First, we give a general definition of post-hoc reversal (Def. 2). If Def. 2 holds with $\boldsymbol{\varphi}_k$'s which are optimal for the base metric $\mathcal{M}_f$, then naive selection becomes suboptimal as it picks $\boldsymbol{\varphi}_k$'s, but $\boldsymbol{\theta}_k$'s are better under the post-hoc metric $\mathcal{M}_{\mathcal{T} \circ f}$. Since the entire space of parameter tuples $\Theta^K$ can be large, we study post-hoc reversal restricted to indexed parameters (Def. 3). Indices can be, for example, training epochs (§ 4.2.1), model sizes (§ 4.2.2) or hyperparameter configurations (§ 4.2.3).

**Definition 2 (Post-hoc reversal)** *Let a post-hoc transform $\mathcal{T}$ map a classifier $f \colon \Theta \times \mathcal{X} \to \mathcal{Y}$ to $\mathcal{T} \circ f \colon \Theta^K \times \mathcal{X} \to \mathcal{Y}$. $\mathcal{T}$ applied to $f$ exhibits post-hoc reversal for a metric $\mathcal{M}$ if there exist $(\boldsymbol{\theta}_1, \ldots, \boldsymbol{\theta}_K), (\boldsymbol{\varphi}_1, \ldots, \boldsymbol{\varphi}_K) \in \Theta^K$ such that $\mathcal{M}_f(\boldsymbol{\theta}_k) \geq \mathcal{M}_f(\boldsymbol{\varphi}_k)$ for all $k = 1, \ldots, K$ but $\mathcal{M}_{\mathcal{T} \circ f}(\boldsymbol{\theta}_1, \ldots, \boldsymbol{\theta}_K) < \mathcal{M}_{\mathcal{T} \circ f}(\boldsymbol{\varphi}_1, \ldots, \boldsymbol{\varphi}_K)$.*

**Definition 3 (Index-wise post-hoc reversal)** *Let $\mathcal{I}$ be a set of indices and $\mathcal{P} \colon \mathcal{I} \to \Theta^K$ map indices to parameter tuples. When Def. 2 holds with $(\boldsymbol{\theta}_1, \ldots, \boldsymbol{\theta}_K) = \mathcal{P}(s), (\boldsymbol{\varphi}_1, \ldots, \boldsymbol{\varphi}_K) = \mathcal{P}(t)$ for some $s, t \in \mathcal{I}$, we call it index-wise post-hoc reversal.*

**Diagnosis.** To enable a visual diagnosis of post-hoc reversal, we define base and post-hoc curves (Def. 4) and a relaxed notion of post-hoc reversal for them (Def. 5). Post-hoc reversal is characterized by non-monotonicity between the base and post-hoc curves, i.e., there exist regions where one improves while the other worsens. This happens, for instance, when one curve exhibits double descent but the other doesn't. Different optimal indices for the two curves is another indicator of post-hoc reversal.

**Definition 4 (Base and post-hoc curves)** *The base and post-hoc curves $\mathcal{M}^{base}, \mathcal{M}^{post} \colon \mathcal{I} \to \mathbb{R}$ are given by $\mathcal{M}^{base}(t) = \frac{1}{K} \sum_{k=1}^{K} \mathcal{M}_f(\boldsymbol{\theta}_k)$ and $\mathcal{M}^{post}(t) = \mathcal{M}_{\mathcal{T} \circ f}(\boldsymbol{\theta}_1, \ldots, \boldsymbol{\theta}_K)$, where $(\boldsymbol{\theta}_1, \ldots, \boldsymbol{\theta}_K) = \mathcal{P}(t)$.*

**Definition 5 (Post-hoc reversal for curves)** *Base and post-hoc curves $\mathcal{M}^{base}, \mathcal{M}^{post} \colon \mathcal{I} \to \mathbb{R}$ exhibit post-hoc reversal when there exist $s, t \in \mathcal{I}$ such that $\mathcal{M}^{base}(s) \geq \mathcal{M}^{base}(t)$ but $\mathcal{M}^{post}(s) < \mathcal{M}^{post}(t)$.*

---

[3]Thus, our variant of SWA is hyperparameter-free.

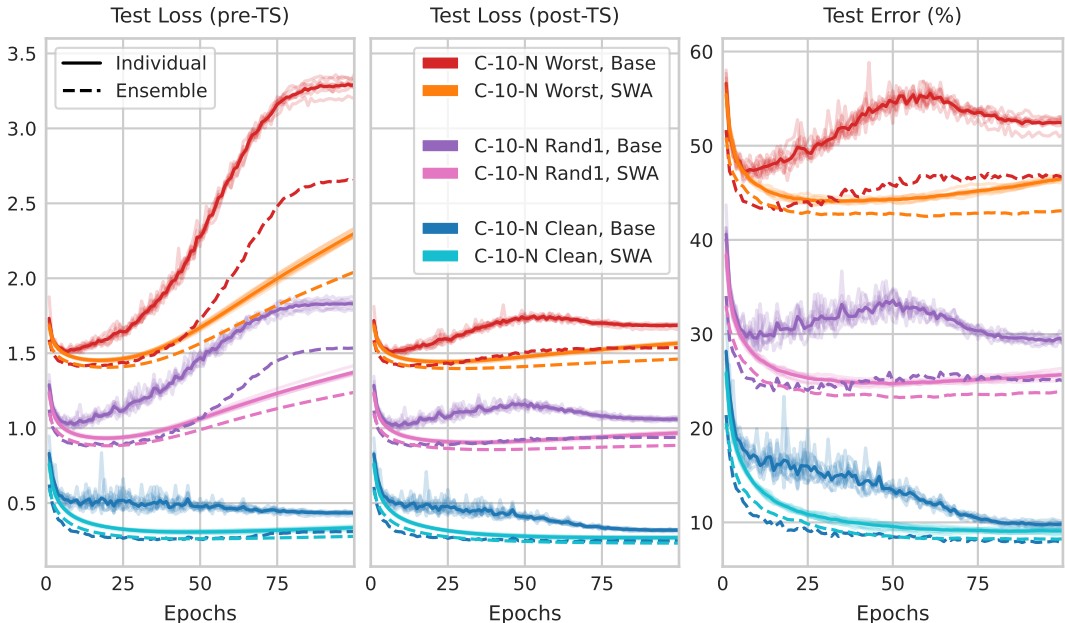

Figure 3: Loss and error for CIFAR-10-N Clean (approx. $0\%$ noise), Rand1 (approx. $17\%$ noise) and Worst (approx. $40\%$ noise). Except for ensemble curves, mean of 8 runs is shown; individual runs are in lighter shades. Ensembles comprise models from these 8 runs. For example, observe post-hoc reversal for C-10-N Worst: (1) error plot: from epoch 5 to 50, solid red (base) curve worsens but solid orange (SWA) curve improves; (2) error plot: solid red (base) curve has a double descent but dashed red (ensemble) curve does not; (3) loss plots: solid red (base) curve has a double descent pre-TS but not post-TS; (4) error plot: best error is at approx. epoch 5 for solid red (base) curve but at approx. epoch 60 for dashed orange (SWA ensemble) curve.

## 4.2 Experiments

### 4.2.1 Epoch-Wise Post-Hoc Reversal

When the indices in Def. 3 are training epochs, we call it *epoch-wise post-hoc reversal*. We use $\theta_t$ to denote the model at the end of epoch $t$. For ensembles, a superscript $j$ denotes the $j$-th training run (out of $N$ runs). $t \in \mathcal{I}$ maps to parameters $\mathcal{P}(t) \in \Theta^K$ ($K = 1$ for TS; $N$ for ensemble; and $t$ for SWA) as follows: $\mathcal{P}_{\text{TS}}(t) = (\theta_t)$; $\mathcal{P}_{\text{Ens}}(t) = (\theta_t^1, \dots, \theta_t^N)^4$.; $\mathcal{P}_{\text{SWA}}(t) = (\theta_1, \dots, \theta_t)$.

**Experimental setup.** We focus on the CIFAR-N dataset [74]. CIFAR-10-N uses the same images as CIFAR-10 but provides multiple human-annotated label sets, allowing the study of realistic noise patterns of varying levels in a controlled manner. Clean is the original label set; Rand1,2,3 are 3 sets of human labels; Aggre combines Rand1,2,3 by majority vote; and Worst combines them by picking an incorrect label, if possible. Similarly CIFAR-100-N has two label sets, Clean and Noisy, with the latter being human-labeled. We train ResNet18 [21] models for 100 epochs with a cosine annealed learning rate. Additional details on datasets and training setup are in App. C. Fig. 3 shows test curves on CIFAR-10-N Clean, Rand1 and Worst. Other label sets and CIFAR-100-N are in App. E. For clarity, we omit the SWA base curve $\mathcal{M}_{\text{SWA}}^{\text{base}}(t) = (\mathcal{M}_f(\theta_1) + \cdots + \mathcal{M}_f(\theta_t))/t$ in the plots, and simply re-use the curve $\mathcal{M}^{\text{base}}(t) = \mathcal{M}_f(\theta_t)$ to compare with the post-hoc SWA curve. While deviating from Def. 4, this better reflects the current practice of early stopping on the latest epoch's base metric.

**Observations.** First, we focus on the base curves: *(1) Overfitting:* As noise increases, test curves go from a single descent to a double descent to a U-shaped curve with increased overfitting. *(2) Double descent:* Noise amplifies double descent, and the second descent worsens with increasing noise (as compared to the first). *(3) Loss-error mismatch:* Loss overfits more drastically than error, leading to a mismatch with higher noise. Optimal models for loss and error can be different.

---

[4]Ensembling models from possibly unequal epochs is covered in § 6

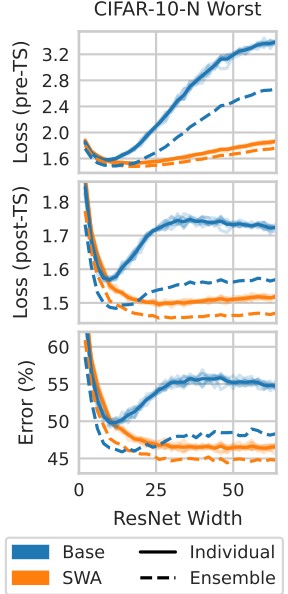

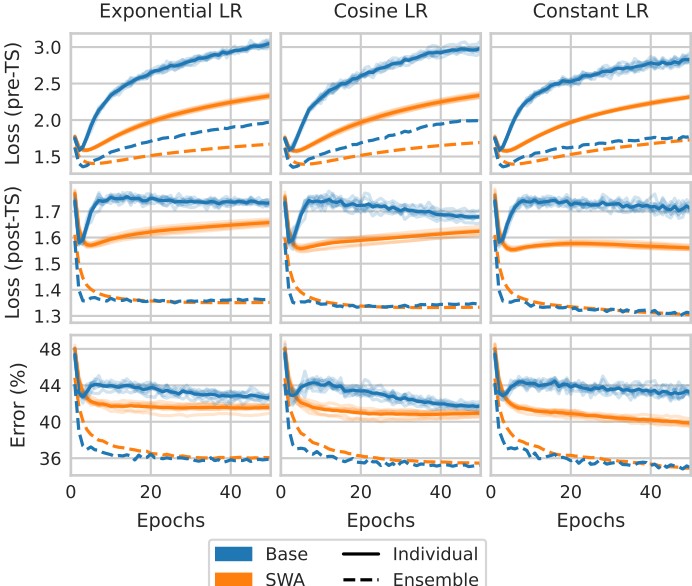

Figure 4: C-10-N Worst test curves against model size. Best width for solid blue curves is $\sim 10$ but for dashed orange curves, it is $\sim 50$ for error and $\sim 25$ for post-TS loss.

Figure 5: FMoW test curves for 3 LR schedules. Note that the pre-TS loss is significantly higher than the post-TS loss. For example, observe post-hoc reversal w.r.t. cosine and constant LRs at epoch 50 between: (1) solid blue (base) and dashed blue (ensemble) error curves; (2) solid blue (base) and solid orange (SWA) post-TS loss curves; (3) solid blue (base) curves for pre-TS and post-TS loss.

Next, we consider the general impact of post-hoc transforms: *(4) Performance improvements:* TS, SWA and ensemble always improve performace, both individually and in composition with larger gaps for noisy label sets. *(5) Post-hoc reversal:* Post-hoc reversal manifests as non-monotonicity between the base and post-hoc curves, especially for noisy label sets. *(6) SWA vs Ensemble:* SWA can recover much of the ensemble gain, but the optimal epoch often differs a lot from the base curve. *(7) Smoother curves:* Base curves fluctuate wildly, but SWA and ensemble curves are smooth, making them more reliable for early stopping.

Finally, we discuss some benefits from post-hoc reversal: *(8) Overfitting:* All transforms reduce overfitting, often reverting performance degradation. *(9) Double descent:* SWA, ensemble and compositions flatten the double descent peak. TS, on the other hand, leads to a double descent for some cases where there was none before. *(10) Loss-error mismatch:* TS aligns the loss and error curves, enabling simultaneously good loss and error.

### 4.2.2 Model-Wise Post-Hoc Reversal

Here, indices represent model sizes. Models of all sizes are trained for $T$ epochs, large enough for convergence. Following [55], we avoid early stopping. Notation-wise, we add a subscript to $\theta$ to indicate the model size $s$. Parameters are indexed as follows: $\mathcal{P}_{\text{TS}}(s) = (\theta_{T,s})$; $\mathcal{P}_{\text{Ens}}(s) = (\theta^1_{T,s}, \ldots, \theta^N_{T,s})$; $\mathcal{P}_{\text{SWA}}(s) = (\theta_{1,s}, \ldots, \theta_{T,s})$.

**Experimental setup.** We parameterize a family of ResNet18s by scaling the number of filters in the convolutional layers. Specifically, we use $[k, 2k, 4k, 8k]$ filters for width $k$. The standard ResNet18 corresponds to $k = 64$. Otherwise the training setup is same as before. Fig. 4 shows the curves. Concretely, the index set $\mathcal{I} = \{2, 4, \ldots, 64\}$ is the set of ResNet widths $k$ described above.

**Observations.** Post-hoc transforms improve performance (up to $\approx 10$ points for error) and mitigate double descent. Further, we see yet another way in which higher-capacity models are better: they give better results under post-hoc transforms even when lower-capacity base models perform better.

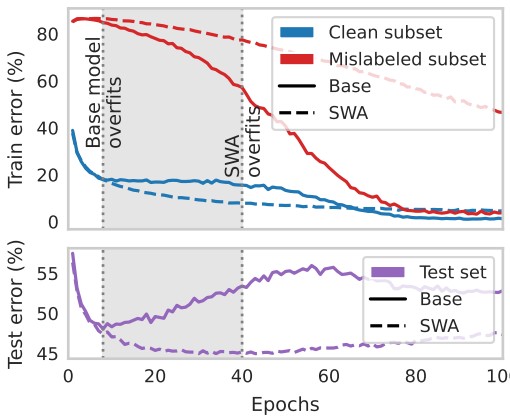

Figure 6: Evolution of the fit/memorization of clean and mislabeled examples during training, for base and SWA models on C-10-N Worst. Train error drops earlier for the clean subset. In the regime of post-hoc reversal (shaded), SWA further lowers the train error on the clean subset, while raising it on the mislabeled subset.

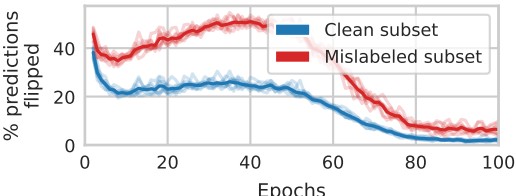

Figure 7: Flipping of predicted class between consecutive epochs, for clean and mislabeled train subsets of C-10-N Worst. % of examples flipped is about twice as high for the mislabeled subset, suggesting an unstable influence on the decision boundary.

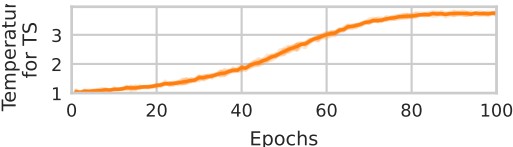

Figure 8: Optimal temperature for TS on C-10-N Worst increases with epochs, indicating increasing overconfidence of the neural network.

#### 4.2.3 Hyperparameter-Wise Post-Hoc Reversal

In general, the index set $\mathcal{I}$ can contain any hyperparameter configurations. Here, we consider two hyperparamters: learning rate schedule and training epochs. To avoid repeating CIFAR-N epoch-wise curves, we experiment on a fresh dataset, FMoW.

**Experimental setup.** We experiment on learning rates (LRs) and training epochs, with index set $\mathcal{I} = \{\texttt{const}, \texttt{exp}, \texttt{cos}\} \times \{1, \ldots, T\}$. Here, $\texttt{const}$, $\texttt{exp}$ and $\texttt{cos}$ refer to constant, exponentially decaying and cosine annealed LRs respectively, and $T$ is the total number of epochs. We train DenseNet121 [26] models on the FMoW dataset [9] which constitutes a 62-way classification of land use from satellite images. For more details, see App. C. Fig. 5 shows the curves.

**LR-wise observations.** We see some interesting instances of post-hoc reversal: (1) constant LR has the worst base performance but the best post-hoc performance; (2) under SWA and TS (composed), the curves continue to improve at the later epochs for constant LR, but not for the decaying LRs[5].

**Epoch-wise observations.** Epoch-wise post-hoc reversal occurs for all LR schedules. SWA and ensembling convert the double descent into a strong single descent, with approx. 10-point improvement in error for the latter. For constant LR, this also changes the optimal epoch. SWA only recovers about half of the ensemble gain, and perhaps surprisingly, ensembling SWA models is not better than ensembling alone. Pre-TS loss curves show a strong mismatch with the error curves, but TS enables simultaneously good loss and error with the last epoch models. Overall, these observations reinforce the trends gleaned from the CIFAR-N experiments.

## 5 Intuitions for Post-Hoc Reversal

In this section, we give hypotheses for post-hoc reversal, backed by experimental evidence.

**Ensembling and SWA delay catastrophic overfitting.** Models learn generalizable patterns from clean examples, and spurious patterns from mislabeled ones. The latter causes overfitting. When noise is low, the former dominates and overfitting is benign. Otherwise, overfitting is catastrophic. Ensembling and SWA improve fitting of clean examples, and reduce memorization of mislabeled ones. When this overturns the dominance of spurious patterns, we observe reversal.

Fig. 6 validates this intuition for SWA on CIFAR-10-N Worst. Fig. 7 further suggests the underlying mechanism — predictions on the mislabeled train subset fluctuate much more during training, allow-

---

[5]Possibly due to higher model variance with constant LR, beneficial for both ensembling and SWA.

Table 1: Naive vs post-hoc (ours) selection for SWA+TS and SWA+Ens+TS transforms. Better values are in bold. Except some clean cases, post-hoc selection is always better, often more than doubling the improvement over no transform. See Tabs. 6 and 8 in App. E for standard deviations.

| Metric → | | Test Loss | | | | Test Error (%) | | | | |
|---|---|---|---|---|---|---|---|---|---|---|
| Transform → | None | SWA+TS | | SWA+Ens+TS | | None | SWA+TS | | SWA+Ens+TS | |
| Dataset ↓ | | Naive | Ours | Naive | Ours | | Naive | Ours | Naive | Ours |
| C-10-N Clean | 0.435 | **0.269** | 0.270 | 0.234 | **0.233** | 9.75 | **9.09** | 9.10 | 8.30 | **8.24** |
| C-10-N Aggre | 0.722 | 0.663 | **0.585** | 0.608 | **0.543** | 19.20 | 17.08 | **16.95** | 15.88 | **15.74** |
| C-10-N Rand1 | 1.009 | 0.968 | **0.907** | 0.916 | **0.859** | 28.63 | 27.13 | **24.84** | 24.80 | **23.50** |
| C-10-N Worst | 1.511 | 1.483 | **1.443** | 1.437 | **1.399** | 46.84 | 46.12 | **44.14** | 44.30 | **42.88** |
| C-100-N Clean | 1.508 | 1.215 | **1.205** | 1.065 | **1.063** | 33.83 | **32.67** | 32.69 | **29.90** | 29.94 |
| C-100-N Noisy | 2.416 | 2.289 | **2.136** | 2.129 | **1.994** | 58.68 | 54.94 | **53.18** | 51.34 | **50.26** |
| FMoW (ID) | 1.583 | 1.627 | **1.554** | 1.494 | **1.305** | 43.20 | 42.69 | **39.92** | 37.95 | **34.93** |
| FMoW (OOD) | 1.831 | 1.840 | **1.788** | 1.700 | **1.571** | 49.32 | 49.70 | **46.75** | 46.74 | **41.56** |

ing SWA to easily revert their memorization. In App. G, we extend this analysis to ensembling and solidify the intuition further by visualizing decision boundaries on a synthetic dataset. This explanation also applies to flattening of the double descent peak, which is a manifestation of catastrophic overfitting.

**TS mitigates loss-error mismatch.** Once a neural net has fit a train example, the cross-entropy loss on it can be lowered by simply upscaling the weights of the linear output layer. This makes the model overconfident later in training, as shown in [19]. For a mislabeled example, this leads to worse loss on similar test instances. The test error is not affected as it is independent of the scale of the logits. In high-noise settings, test loss can worsen due to memorization of mislabeled examples, even as the test error improves from continued learning on clean examples, leading to loss-error mismatch. TS fixes this by downscaling the logits. Indeed, one finds that the temperature (as obtained with a held-out set) increases with epochs (Fig. 8).

**Post-hoc reversal can occur against epochs, model sizes or other hyperparameters.** Different variants of post-hoc reversal can be unified via *effective model complexity* (EMC), introduced in [55] to unify epoch- and model-wise double descent. EMC measures memorization capacity, which plays a key role in post-hoc reversal. EMC increases with epochs and model size. Further, EMC increases with epochs more rapidly for constant LR than annealed LR, explaining our observations in § 4.2.3.

# 6 Post-Hoc Selection: Leveraging Post-Hoc Reversal in Practice

Our findings from §4 motivate the principle of *post-hoc selection*, where model development decisions take post-hoc transforms into account. For concreteness, we discuss the choice of checkpoints from training runs under the SWA+TS and SWA+Ens+TS transforms. Checkpoint selection reduces to the selection of the final epoch $\widehat{T}$, as SWA uses all checkpoints up to that epoch. $\mathcal{M}_{\text{val}}$ denotes a metric of choice computed on the validation set.

**SWA+TS.** Naive selection picks epoch $\widehat{T} = \arg\min_T \mathcal{M}_f^{\text{val}}(\theta_T)$. In contrast, post-hoc selection picks $\widehat{T} = \arg\min_T \mathcal{M}_{\mathcal{T}_{\text{S+T}} \circ f}^{\text{val}}((\theta_t)_{t=1}^T)$.

**SWA+Ens+TS.** Here we have $N$ different training runs to pick epochs for. Naive selection picks $\widehat{T}_j = \arg\min_T \mathcal{M}_f^{\text{val}}(\theta_T^j)$ for each run independently. In contrast, post-hoc selection would ideally pick $\widehat{T}_1, \ldots, \widehat{T}_N = \arg\min_{T_1, \ldots, T_N} \mathcal{M}_{\mathcal{T}_{\text{S+E+T}} \circ f}^{\text{val}}((\theta_t^1)_{t=1}^{T_1}, \ldots, (\theta_t^N)_{t=1}^{T_N})$ which jointly minimizes the ensemble performance. This being computationally expensive, we instead minimize under the constraint $\widehat{T}_1 = \cdots = \widehat{T}_N$[6]

**Results.** Tab. 1 compares naive and post-hoc selection strategies for CIFAR-N and FMoW. Except for some clean label sets, post-hoc selection is always better than naive selection, often with $> 2\times$ improvement from post-hoc selection as compared to naive selection. It remains effective with out-

---

[6]Alternatively, one can select $\widehat{T}_j = \arg\min_T \mathcal{M}_{\mathcal{T}_{\text{S+T}} \circ f}^{\text{val}}(\theta_1^j, \ldots, \theta_T^j)$ as a hybrid between post-hoc selection (within runs) and naive selection (across runs).

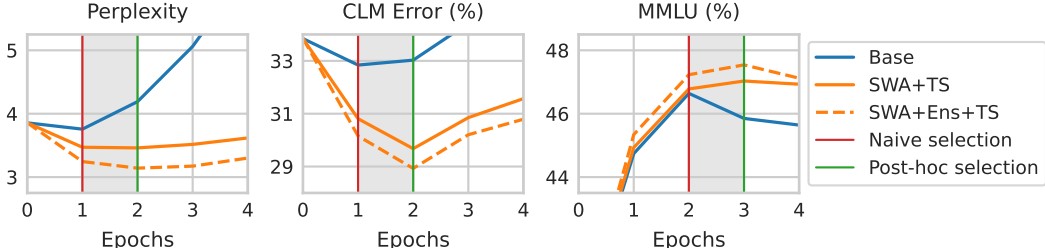

Figure 9: Perplexity and causal language modeling (CLM) error on the Guanaco test set, and MMLU accuracy (higher is better) for instruction tuning LLaMA-2-7B. Shading indicates post-hoc reversal. Base and SWA+TS curves are mean of 8 runs; SWA+Ens+TS ensembles models from these runs. Individual runs are not shown as they have high variance (see Tab. 7 in App. E).

of-distribution (OOD) val/test sets, as seen for FMoW (we use ID and OOD splits from WILDS [34]). For some datasets, like C-100-N Noisy, post-hoc selection is only marginally better on test error. Often, in such cases, the error floor is already quite high (e.g., C-100-N Noisy has $\sim 40\%$ noise and ResNet-18 has $\sim 10\%$ error on clean C-100, so a test error of $\sim 50\%$ is already impressive), and test loss is a more appropriate metric.

**Early stopping.** We advocate monitoring post-hoc metrics for early stopping. Only a running average needs to be updated for SWA, and TS involves a quick single-parameter optimization. Further, while the base curves can fluctuate wildly between consecutive runs, SWA+TS curves are considerably smoother (see Figs. 3, 11 and 10), making them more reliable for automated early stopping. One can similarly monitor metrics for SWA+Ens+TS under parallel training runs.

## 7    Experiments Across Domains and Modalities

In § 4 and § 6, we introduced post-hoc reversal and selection with experiments on the CIFAR-N and FMoW datasets. In this section, we supplement our experimental analysis with additional experiments across diverse domains and modalities to demonstrate the generality of our findings.

### 7.1    LLM Instruction Tuning

Language models are pre-trained or fine-tuned with a self-supervised objective of predicting the next token in a text corpus. There might be many acceptable tokens following a given prefix, albeit with different probabilities. Thus next token prediction is noisy and one might reasonably expect to see post-hoc reversal. In this section, we test this hypothesis for the task of fine-tuning LLMs to follow instructions (instruction tuning [72]). Instruction tuning datasets are naturally small [85] and amenable to multi-epoch training where catastrophic overfitting becomes an important concern. Recent works [53, 81] have argued for data repetitions for LLM pre-training as well, but such experiments are beyond the scope of this paper.

**Experimental setup.** We fine-tune LLaMA-2-7B [70] on the Guanaco dataset [12] of chat completions. We evaluate perplexity and causal language modeling (CLM) error on the test set, and also the MMLU accuracy [24] to better contextualize model improvements. Fig. 9 shows the curves. Tab. 7 in App. E gives exact numbers, and App. F explores sub-epoch checkpointing. For TS, we use a shared temperature parameter to scale the logits of all tokens and leave more involved strategies like *long-horizon temperature scaling* [66] to future work.

**Observations.** We observe post-hoc reversal between epochs 1 and 2 for perplexity and error, and between epochs 2 and 3 for MMLU. Both SWA+TS and SWA+Ens+TS transforms show significant improvements, much of which is only realized under post-hoc selection.

### 7.2    Other Text, Tabular and Graph Datasets

In this section, we further expand our experimental coverage to text, tabular and graph classification datasets from real-world applications.

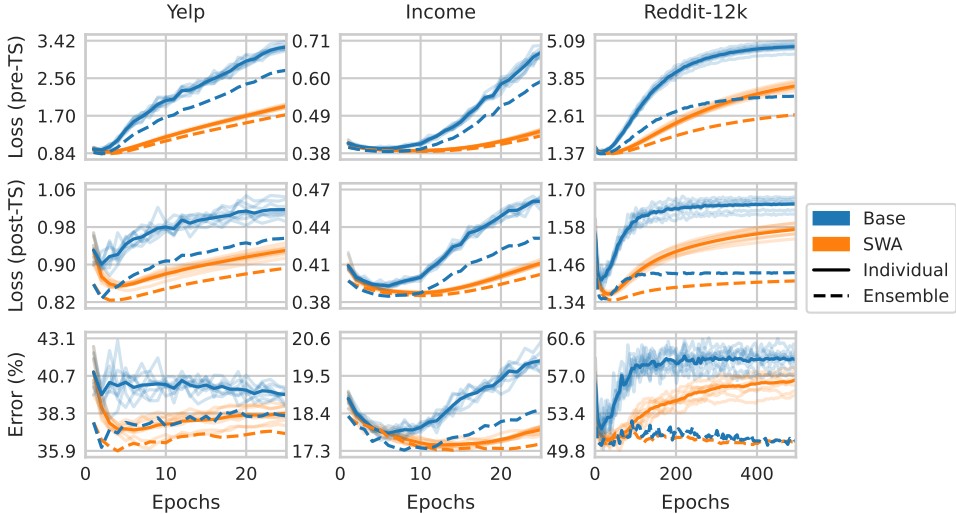

Figure 10: Test curves for 3 real-world noisy datasets. Note that the pre-TS loss is significantly higher than the post-TS loss. Examples of post-hoc reversal between the base curves given by the solid blue lines and the post-hoc curves given by the dashed orange lines (SWA ensemble): (1) optimal epoch is different for base and post-hoc curves for error and post-TS loss on all datasets; (2) for error on Yelp, base curve shows double descent but post-hoc curve does not; (3) for error on Income, base curve overfits catastrophically at approx. epoch 5 but post-hoc curve continues improving till approx. epoch 20; (4) for error on Reddit-12k, base curve does not show double descent but post-hoc curve does.

**Experimental setup.** We consider the following tasks: (1) sentiment classification on the Yelp reviews dataset [5] (text) with a pre-trained transformer BERT [13], (2) prediction tasks on census data from Folktables [14] (tabular) with MLPs and (3) community detection on the Reddit and Collab datasets [82] (graph) with graph neural networks (GNNs). Folktables has 5 prediction tasks: Income, PublicCoverage, Mobility, Employment and TravelTime. Reddit has 2 versions: Reddit-5k and Reddit-12k. For more details, see App. C. Figure 10 shows curves for Yelp, Income and Reddit-12k. Tab. 5 in App. D compares naive and post-hoc selection on all datasets.

**Observations.** Post-hoc reversal is a recurring feature across datasets, transforms and metrics. The 3 datasets show different patterns between the base and post-hoc curves, showing that post-hoc reversal can take a variety of forms.

# 8 Conclusion

We empirically studied temperature scaling (TS), ensembling, stochastic weight averaging (SWA) and their compositions, and found that these transforms can reverse model peformance trends (post-hoc reversal). Based on our findings, we presented the simple technique of post-hoc selection, and showed that it outperforms naive selection. We validated our findings and proposals over diverse settings.

Our work has broad implications for the field of deep learning. It shows that current practices surrounding the use of post-hoc transforms leave much room for improvement. This is especially true for noisy data, which is pervasive in real-world applications. Future directions include better strategies for checkpoint selection, developing a theoretical understanding, investigating impacts on scaling laws, and characterizing other instances of post-hoc reversal.

**Summary of practical recommendations.** We advocate for the use of TS, ensembling and SWA across deep learning applications. Further, such transforms should be tightly integrated into the model development pipeline, following the methodology outlined in the paper. In particular: (1) apply SWA+TS and SWA+Ens+TS transforms for better results in the single- and multi-model settings respectively; (2) track temperature-scaled loss to overcome loss-error mismatch; (3) monitor post-hoc metrics to avoid premature early stopping; (4) make hyperparameter decisions informed by post-transform performance; (5) use post-hoc selection to pick model checkpoints.

**Acknowledgements**

ZL acknowledges Amazon AI, Salesforce Research, Facebook, UPMC, Abridge, the PwC Center, the Block Center, the Center for Machine Learning and Health, and the CMU Software Engineering Institute (SEI) via Department of Defense contract FA8702-15-D-0002, for their generous support of ACMI Lab's research on machine learning under distribution shift.

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

# A  Expanded Related Work

**Phenomena.** Empirical works like double descent [55], grokking [58], scaling laws [31], neural-collapse [57], edge-of-stability [10], lottery-ticket-hypothesis [17] have revealed both challenges and oppotunities for improving the understanding and practices of deep neural network training. Post-hoc reversal expands this list as a novel phenomenon regarding learning dynamics under the lens of post-hoc transforms. It is most intimately connected with double descent, offering a way to mitigate it. Some works [7, 23, 54, 59, 65, 76] show other mitigations, such as regularization and data augmentation.

**Temperature Scaling (TS).** TS belongs to a family of post-hoc calibration techniques [2, 19, 32, 66, 83], with the unique property of preserving classification error. Recently, calibration has been applied to large vision and language models [11, 71, 84]. While loss-error mismatch has been reported before [11, 19], to the best of our knowledge, we are the first to report post-hoc reversal with TS.

**Ensembling.** Ensembling is a foundational technique in machine learning, encompassing bagging, boosting, etc. In deep learning, a uniform ensemble is most popular [3, 39], although recent work on ensembling LLMs has explored more efficient routing-based ensembles [29, 46, 48, 49]. Various works have explored strategies to form optimal ensembles [36, 47, 51, 77], generally based on model diversity [38], but recently Abe et al. [1] have warned against this. In contrast, our recommendation for forming ensembles relies directly on the validation performance of the ensemble, introducing no proxies, and still being computationally cheap.

**Stochastic Weight Averaging (SWA).** SWA [28] is the culmination of a line of work [18, 25] which seek to cheaply approximate ensembling. It has inspired numerous works which average weights in some form [4, 6, 27, 41, 61, 77] often in combination with ensembling. Recently, weight averaging has shown up in the LLM space [62, 64]. While these works generally apply SWA with a fixed training time determined independently, we present SWA in the role of early stopping and model selection. In practice, SWA has often been found to be unreliable[7], and is often skipped from training recipes even when considered [35, 75]. Our work sheds some light on this, offering a rather counter-intuitive choice of models to include in the weight average for best results.

**Noise.** Many training strategies have been introduced to deal with noisy data (see [69] for a survey). However, the efficacy of simple post-hoc transforms has been left unexplored. Further, most of these works are motivated by labeling errors, which leaves some of the core practical considerations for dealing with general noisy data unaddressed. For instance, access to a clean validation set is assumed and test loss is overlooked as an important metric [43, 44]. We also entirely avoid experiments on synthetic noise, informed by recent work which questions the transferability of findings to realistic noise patterns [30, 73]. Some recent datasets [30, 40, 68, 73, 79] make it possible to study realistic noise along with known noise estimates. Noise due to insufficient information in the input context (Fig. 11) has also been studied under different settings, such as for RLHF [67].

**Multi-epoch training of LLMs.** Multi-epoch training of LLMs runs into severe catastrophic overfitting. Xue et al. [81] examine the contributing factors and explore possible solutions. They find that regularization is not helpful, except for dropout. Muennighoff et al. [53] study scaling laws considering data repetitions. Complementarily, we put forward post-hoc transforms as an effective solution with our post-hoc selection methodology. This is especially important for fine-tuning LLMs, e.g. in instruction tuning [72], where [85] and [8] advocate for fine-tuning with a smaller amount of higher quality samples for more epochs.

# B  Expanded Preliminaries and Background

## B.1  Learning on Noisy Data

Figures 12, 11 and 13 illustrate various sources of noise: aleatoric unertainty, epistemic uncertainty and annotation errors. Below we provide some background on Bayes-optimal classifier and use it to introduce the clean error metric and Bayes loss/error as measures of noise level.

---

[7]See, for example, discussion at `https://discuss.huggingface.co/t/improvements-with-swa/858`.

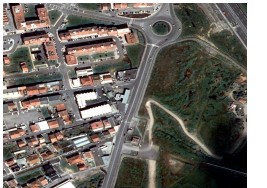 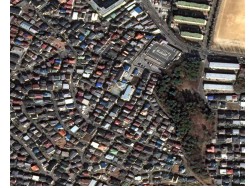

(a) *Gas Station*   (b) *Educational Institution*

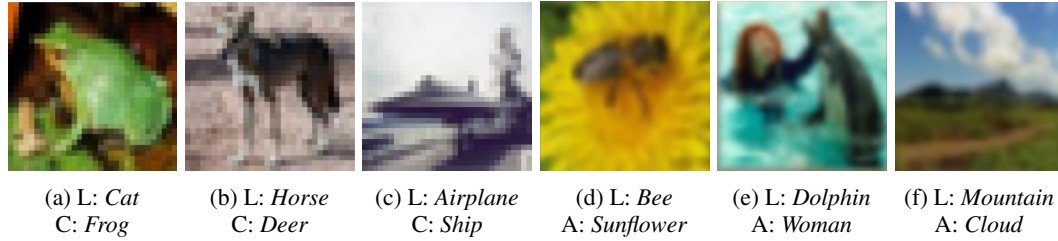

```
Write a ballad on why internet privacy and open-source is...
├── There once waw a computer, a friendly old computer,...
│   └── You said waw instead of was. Not sure the grammar...
│       ├── Sorry, I did indeed misspell "was". here is a...
│       ├── There once was a computer, a friendly old...
│       └── There once was a computer, friendly and old,...
├── Verse 1 Open source and privacy, We need to protect our...
│   └── Looks good but can you explain who takes away the...
└── Verse 1: In a world of endless clicks and endless...
    ├── Can you modify it to carry an opposite message?
    └── Change the chorus
        ├── Chorus: Oh internet, let freedom be our guide...
        ├── Chorus: Oh internet, where information flows...
        └── New Chorus: Oh internet, where knowledge and...
```

Figure 11: Data can be noisy due to insufficient information in the input context (epistemic uncertainty). Figures 11a and 11b show satellite images from the FMoW dataset. The labels are correct, as corroborated by external map data. However, they cannot be determined with full certainty from the images alone.

Figure 12: Data can be noisy due to non-determinism in the prediction target (aleatoric uncertainty). Figure shows a message tree from the OpenAssistant Conversations (OASST1) Dataset. A chatbot can continue a conversation satisfactorily in many different ways, making next token prediction noisy.

| (a) L: *Cat* C: *Frog* | (b) L: *Horse* C: *Deer* | (c) L: *Airplane* C: *Ship* | (d) L: *Bee* A: *Sunflower* | (e) L: *Dolphin* A: *Woman* | (f) L: *Mountain* A: *Cloud* |

Figure 13: Data can be noisy due to annotation errors. Figures 13a, 13b and 13c are mislabeled images from CIFAR-10. 13d, 13e and 13f are ambiguous images from CIFAR-100 with multiple correct labels among the given classes. (**L** = label in dataset, **C** = correct label, **A** = alternative label)

**Bayes-optimal classifier.** $f_{\mathcal{D}}$, given by $f_{\mathcal{D}}(\mathbf{x})_k = \log \mathcal{P}_{\mathcal{D}}[y = k \mid \mathbf{x}]$ minimizes both $\mathcal{M}_{\mathcal{D}}^{\text{error}}$ and $\mathcal{M}_{\mathcal{D}}^{\text{loss}}$, and is called the *Bayes-optimal classifier* for $\mathcal{D}$. The *Bayes error* $\mathcal{M}_{\mathcal{D}}^{\text{error}}[f_{\mathcal{D}}]$ and *Bayes loss* $\mathcal{M}_{\mathcal{D}}^{\text{loss}}[f_{\mathcal{D}}]$ are measures of the noise level. $y^*(\mathbf{x}) = \arg\max_k f_{\mathcal{D}}(\mathbf{x})_k$ is sometimes called the *clean label*. Using $y^*$, one may define the *clean data distribution* $\widetilde{\mathcal{D}}$ with $\mathcal{P}_{\widetilde{\mathcal{D}}}[\mathbf{x}] = \mathcal{P}_{\mathcal{D}}[\mathbf{x}]$ and $\mathcal{P}_{\widetilde{\mathcal{D}}}[y \mid \mathbf{x}] = \mathbf{1}\{y = y^*(\mathbf{x})\}$. The *clean error* $\mathcal{M}_{\widetilde{\mathcal{D}}}^{\text{error}}$ is a common metric in the label noise literature but not a focus of our work as $y^*$ is typically inaccessible in more general noisy settings.

### B.2 Post-Hoc Transforms in Machine Learning

The explicit forms of the composed transforms SWA+TS and SWA+Ens+TS (denoted as $\mathcal{T}_{\text{S+T}}$ and $\mathcal{T}_{\text{S+E+T}}$) are given by Equations 4 and 5 respectively. For $\mathcal{T}_{\text{S+E+T}}$, parameters $\theta_1^l, \ldots, \theta_{K_l}^l$ are weight-averaged and the $L$ resulting models are ensembled, followed by temperature scaling. $\tau_l$ is the temperature for weight-averaged models, and $\tau_{\text{Ens}}$ is the temperature for the ensemble. As before, they are obtained by optimizing the cross-entropy loss over the validation set, with model parameters fixed.

$$(\mathcal{T}_{\text{S+T}} \circ f)(\mathbf{x}; \theta_1, \ldots, \theta_K) = \frac{1}{\tau} f\left(\mathbf{x}; \frac{1}{K}\sum_{i=1}^{K}\theta_i\right), \text{ with } \tau = \arg\min_{\tau} \mathcal{M}_{\text{val}}^{\text{loss}}\left[\frac{1}{\tau}f\left(\cdot; \frac{1}{K}\sum_{i=1}^{K}\theta_i\right)\right]$$
(4)

$$(\mathcal{T}_{\text{S+E+T}} \circ f)\left(\mathbf{x}; \theta_1^1, \ldots, \theta_{K_1}^1, \ldots, \theta_1^L, \ldots, \theta_{K_L}^L\right) = \frac{1}{\tau_{\text{Ens}}} \frac{1}{L}\sum_{l=1}^{L}\frac{1}{\tau_l}f\left(\mathbf{x}; \frac{1}{K_l}\sum_{k=1}^{K_l}\theta_k^l\right)$$
(5)

Table 2: Dataset Details.

| Modality | Dataset | Train Size | Val Size | Test Size | Classes | Input Size | Units |
|---|---|---|---|---|---|---|---|
| Vision | CIFAR-10 | 40000 | 5000 | 5000 | 10 | $3 \times 32 \times 32$ | |
| | CIFAR-100-N Coarse | 40000 | 5000 | 5000 | 20 | $3 \times 32 \times 32$ | $C \times W \times H$ |
| | CIFAR-100-N Fine | 40000 | 5000 | 5000 | 100 | $3 \times 32 \times 32$ | |
| | FMoW | 76863 | 11483 | 11327 | 62 | $3 \times 224 \times 224$ | |
| Text | Guanaco | 8850 | 500 | 500 | 32000 | $\lesssim 4000$ | characters |
| | Yelp | 25000 | 5000 | 5000 | 5 | $\lesssim 2000$ | |
| Tabular | Income | 156533 | 19566 | 19566 | 2 | 816 | |
| | Public Coverage | 110844 | 13855 | 13855 | 2 | 88 | |
| | Mobility | 64265 | 8032 | 8032 | 2 | 101 | features |
| | Employment | 303055 | 37881 | 37881 | 2 | 98 | |
| | Travel Time | 138008 | 17250 | 17250 | 2 | 615 | |
| Graph | Collab | 4000 | 500 | 500 | 3 | 74.49, 2457.78 | |
| | Reddit-5k | 4001 | 499 | 499 | 5 | 508.52, 594.87 | nodes, edges (avg.) |
| | Reddit-12k | 9545 | 1192 | 1192 | 11 | 391.41, 456.89 | |

Table 3: Training Details.

| Dataset | Model | Pre-train | Optimizer | LR | Weight Decay | LR Schedule | Epochs | Batch Size |
|---|---|---|---|---|---|---|---|---|
| C-10/100-N | ResNet18-D [22] | Yes | SGD | 0.1 | 5e-4 | Cosine | 100 | 500 |
| FMoW | DenseNet121 [26] | Yes | Adam | 1e-4 | 0 | Constant | 50 | 64 |
| Guanaco | LLaMA-2-7B [70] | Yes | Adam | 2e-4 | 0 | Constant | 6 | 16 |
| Yelp | BERT [13] | Yes | AdamW | 5e-5 | 1e-2 | Linear | 25 | 16 |
| Folktables | MLP | No | Adam | 0.01 | 0 | Exponential | 50 | 256 |
| Collab | GIN [80] | No | Adam | 0.01 | 0 | Exponential | 500 | 128 |
| Reddit | GCN [33] | No | Adam | 0.01 | 0 | Exponential | 500 | 128 |

## C  Dataset and Training Details

Tabs. 2 and 3 summarize the datasets and training details for our experiments. They are described in detail below. We trained our models under these hyperparameters on 48 GB A6000 GPUs in a single-GPU setup, except for LLaMA-2-7B fine-tuning on Guanaco, for which we used 80 GB A100 GPUs. Single model training completes in a few hours for all datasets except FMoW and Guanaco, on which training took upto 12 hours. We experiment most extensively on the CIFAR-N datasets, where our optimized script can train a single model in 3-5 minutes on an A6000 GPU.

**CIFAR-N [74].** CIFAR-10-N uses the same images as CIFAR-10 but provides multiple human-annotated label sets. Clean is the original label set; Rand1,2,3 are 3 sets of human labels; Aggre combines Rand1,2,3 by majority vote; and Worst combines them by picking an incorrect label, if possible. CIFAR-100 has 2 variants, a fine-grained one with 100 classes and a coarse-grained one with 20 classes, obtained by grouping the fine-grained classes. Correspondingly, there are CIFAR-100-N Coarse and CIFAR-100-N Fine datasets. They have two label sets each: Clean and Noisy, with the latter being human-labeled. In the main paper, CIFAR-100-N refers to the fine-grained version.

By cross-referencing with the original labels, it is possible to estimate the noise levels. These are shown in Table 4.

CIFAR-N allows access to clean labels. In the literature, the validation and test sets for CIFAR-N typically use the clean labels [42, 45, 78]. However, access to clean labels is a luxury only available for label noise settings. Even there, obtaining clean labels is expensive, as it requires careful expert annotation. For other sources of noise it might not even be feasible to obtain clean labels. Hence, we restrict ourselves to using noisy (*i.i.d.* to train) validation and test sets. Since CIFAR-N only provides

Table 4: Noise levels for CIFAR-N (%), reproduced from [74].

| | CIFAR-10-N | | | | | | CIFAR-100-N Coarse | | CIFAR-100-N Fine | |
|---|---|---|---|---|---|---|---|---|---|---|
| Clean | Aggre | Rand1 | Rand2 | Rand3 | Worst | Clean | Noisy | Clean | Noisy |
| 0.00 | 9.03 | 17.23 | 18.12 | 17.64 | 40.21 | 0.00 | 25.60 | 0.00 | 40.20 |

Table 5: Naive vs post-hoc (ours) selection for SWA+TS and SWA+Ens+TS transforms on some real-world datasets. Better values are in bold.

| Metric → | Test Loss | | | | | Test Error (%) | | | | |
| Transform → | None | SWA+TS | | SWA+Ens+TS | | None | SWA+TS | | SWA+Ens+TS | |
| Dataset ↓ | | Naive | Ours | Naive | Ours | | Naive | Ours | Naive | Ours |
|---|---|---|---|---|---|---|---|---|---|---|
| Yelp | 0.908 | 0.890 | **0.854** | 0.841 | **0.824** | 39.41 | 38.02 | **37.33** | 36.18 | **36.14** |
| Income | 0.393 | 0.390 | **0.387** | 0.388 | **0.385** | 17.84 | 17.69 | **17.54** | 17.62 | **17.40** |
| PublicCoverage | 0.544 | 0.540 | **0.539** | 0.538 | **0.538** | 27.52 | 27.31 | **27.25** | 27.25 | **27.02** |
| Mobility | 0.474 | 0.472 | **0.471** | 0.471 | **0.468** | 21.43 | **21.38** | 21.42 | **21.17** | 21.24 |
| Employment | 0.380 | 0.379 | **0.378** | 0.378 | **0.377** | 17.94 | **17.77** | 17.80 | **17.72** | 17.83 |
| TravelTime | 0.597 | 0.597 | **0.593** | 0.596 | **0.591** | 35.77 | 35.46 | **35.35** | 35.44 | **35.23** |
| Collab | 0.492 | 0.475 | **0.460** | 0.439 | **0.404** | 20.65 | 21.58 | **20.27** | 20.40 | **18.80** |
| Reddit-5k | 1.154 | 1.112 | **1.100** | 1.101 | **1.085** | 47.42 | 48.35 | **47.04** | 47.09 | **45.49** |
| Reddit-12k | 1.405 | 1.381 | **1.366** | 1.367 | **1.346** | 51.78 | **51.08** | 51.11 | **50.34** | 51.26 |

human labels for the original 50k CIFAR-10/100 train images, we split these into 40k/5k/5k images for train/val/test sets.

**FMoW [9, 34].** This is the version of the original FMoW dataset [9] as used in the WILDS benchmark [34]. For FMoW (ID) we use the in-distribution val and test sets, and for FMoW (OOD), we use the out-of-distribution val and test sets, where the val set is shifted with respect to the train set, and the test set is shifted with respect to both the train and val sets. All splits are as provided by WILDS. The input is an RGB satellite image (rescaled to 224 x 224 pixels) and the label is one of 62 building or land use categories. The labels were obtained by a combination of human annotation and cross-referenced geographical information. The original dataset provides additional metadata about location, time, sun angles, physical sizes, etc. which is ignored in the WILDS dataset (and hence in ours). While the labels have low noise compared to the ground-truth, this dataset is noisy because of insufficient information. It is hard to disambiguate the building or land use category with full certainty by looking at the satellite image alone. See Figure 11. Models and training setup are as used in [9, 34], except for the LR schedule, where we experiment with multiple alternatives.

**Guanaco [12].** This is a subset of the OASST1 dataset [37] containing only the highest-rated paths in the conversation tree. We follow the fine-tuning setup from [12], except that we use vanilla fine-tuning without any quantization or low-rank adapters.

**Yelp [5].** This is a subset of the Yelp Dataset Challenge 2015 dataset with 25k reviews in the train set and 5k reviews each in the validation and test sets. The input is a review text and the label is one of 5 classes (1 to 5 stars). Assigning a rating to a review is intrinsically non-deterministic as different reviewers might have different thresholds for the star ratings. This introduces noise in the data.

**Folktables [14].** Folktables consists of 5 classification tasks based on the US Census: Income, Employment, Health, TravelTime and PublicCoverage. The data is tabular. The available feature columns do not contain sufficient information to predict the targets with full certainty, even if the Census recorded the ground-truth labels with high accuracy. This results in noise.

**Collab and Reddit [52, 82].** These datasets are from TUDataset [52], and were originally introduced by Yanardag and Vishwanathan [82]. Collab is a scientific collaboration dataset. The input is an ego-network of a researcher and the label is the field of the researcher (one of High Energy Physics, Condensed Matter Physics and Astro Physics). The Reddit-5k and Reddit-12k datasets (originally called REDDIT-MULTI-5K and REDDIT-MULTI-12K) are balanced datasets where the input is a graph which corresponds to an online discussion thread from the social network site Reddit. Nodes correspond to users and there is an edge if one user responded to another's comment. The task is to predict which subreddit a discussion graph belongs to. Reddit-5k is smaller with 5k examples and 5 classes. Reddit-12k is bigger with 12k examples and 11 classes.

Table 6: Detailed results for CIFAR-N datasets. **Base** denotes no transform and **Final** denotes the SWA+Ens+TS transform. **Gain** shows performance improvement. $\Delta$ shows change from naive selection to post-hoc selection. Since Base and Gain columns involve $8$ individual runs, we report $\text{mean}_{\pm\text{std. dev.}}$ of the metric. C-10-N, C-100-N-C and C-100-N-F are shorthands for CIFAR-10-N, CIFAR-100-N Coarse and CIFAR-100-N Fine respectively.

| Metric → | | | Test Loss | | | | Test Error (%) | | |
|---|---|---|---|---|---|---|---|---|---|
| Dataset ↓ | Select ↓ | Epochs | Base | Final | Gain | Epochs | Base | Final | Gain |
| C-10-N Clean | Naive | $90_{\pm 9}$ | $0.435_{\pm 0.012}$ | 0.234 | $0.201_{\pm 0.012}$ | $92_{\pm 5}$ | $9.75_{\pm 0.24}$ | 8.30 | $1.45_{\pm 0.24}$ |
| | Post-hoc | 100 | $0.433_{\pm 0.009}$ | 0.233 | $0.200_{\pm 0.009}$ | 96 | $9.82_{\pm 0.27}$ | 8.24 | $1.58_{\pm 0.27}$ |
| | $\Delta$ | ↑ $10_{\pm 9}$ | ↓ $0.001_{\pm 0.005}$ | ↓ 0.001 | – | ↑ $4_{\pm 5}$ | ↑ $0.07_{\pm 0.10}$ | ↓ 0.06 | – |
| C-10-N Aggre | Naive | $10_{\pm 3}$ | $0.722_{\pm 0.018}$ | 0.608 | $0.114_{\pm 0.018}$ | $94_{\pm 6}$ | $19.20_{\pm 0.39}$ | 15.88 | $3.33_{\pm 0.39}$ |
| | Post-hoc | 53 | $0.977_{\pm 0.030}$ | 0.543 | $0.434_{\pm 0.030}$ | 58 | $22.21_{\pm 0.62}$ | 15.74 | $6.47_{\pm 0.62}$ |
| | $\Delta$ | ↑ $43_{\pm 3}$ | ↑ $0.255_{\pm 0.027}$ | ↓ 0.065 | – | ↓ $36_{\pm 6}$ | ↑ $3.00_{\pm 0.73}$ | ↓ 0.14 | – |
| C-10-N Rand1 | Naive | $8_{\pm 2}$ | $1.009_{\pm 0.008}$ | 0.916 | $0.093_{\pm 0.008}$ | $22_{\pm 32}$ | $28.63_{\pm 0.57}$ | 24.80 | $3.83_{\pm 0.57}$ |
| | Post-hoc | 31 | $1.189_{\pm 0.017}$ | 0.859 | $0.330_{\pm 0.017}$ | 67 | $31.58_{\pm 0.51}$ | 23.50 | $8.08_{\pm 0.51}$ |
| | $\Delta$ | ↑ $23_{\pm 2}$ | ↑ $0.181_{\pm 0.018}$ | ↓ 0.057 | – | ↑ $44_{\pm 32}$ | ↑ $2.95_{\pm 0.95}$ | ↓ 1.30 | – |
| C-10-N Rand2 | Naive | $10_{\pm 1}$ | $1.040_{\pm 0.008}$ | 0.931 | $0.108_{\pm 0.008}$ | $14_{\pm 6}$ | $29.90_{\pm 0.42}$ | 25.44 | $4.47_{\pm 0.42}$ |
| | Post-hoc | 30 | $1.189_{\pm 0.037}$ | 0.888 | $0.301_{\pm 0.037}$ | 74 | $31.15_{\pm 0.38}$ | 24.12 | $7.02_{\pm 0.38}$ |
| | $\Delta$ | ↑ $20_{\pm 1}$ | ↑ $0.150_{\pm 0.038}$ | ↓ 0.043 | – | ↑ $60_{\pm 6}$ | ↑ $1.24_{\pm 0.56}$ | ↓ 1.32 | – |
| C-10-N Rand3 | Naive | $9_{\pm 2}$ | $1.005_{\pm 0.014}$ | 0.910 | $0.095_{\pm 0.014}$ | $24_{\pm 30}$ | $28.96_{\pm 0.65}$ | 24.86 | $4.10_{\pm 0.65}$ |
| | Post-hoc | 32 | $1.179_{\pm 0.027}$ | 0.864 | $0.315_{\pm 0.027}$ | 38 | $32.39_{\pm 0.68}$ | 23.44 | $8.95_{\pm 0.68}$ |
| | $\Delta$ | ↑ $23_{\pm 2}$ | ↑ $0.174_{\pm 0.031}$ | ↓ 0.046 | – | ↑ $14_{\pm 30}$ | ↑ $3.43_{\pm 1.03}$ | ↓ 1.42 | – |
| C-10-N Worst | Naive | $8_{\pm 2}$ | $1.511_{\pm 0.008}$ | 1.437 | $0.073_{\pm 0.008}$ | $10_{\pm 3}$ | $46.84_{\pm 0.56}$ | 44.30 | $2.54_{\pm 0.56}$ |
| | Post-hoc | 25 | $1.643_{\pm 0.019}$ | 1.399 | $0.245_{\pm 0.019}$ | 24 | $49.67_{\pm 0.74}$ | 42.88 | $6.79_{\pm 0.74}$ |
| | $\Delta$ | ↑ $17_{\pm 2}$ | ↑ $0.133_{\pm 0.018}$ | ↓ 0.039 | – | ↑ $14_{\pm 3}$ | ↑ $2.83_{\pm 0.94}$ | ↓ 1.42 | – |
| C-100-N-C Clean | Naive | $33_{\pm 35}$ | $1.011_{\pm 0.014}$ | 0.669 | $0.342_{\pm 0.014}$ | $91_{\pm 4}$ | $23.12_{\pm 0.40}$ | 19.36 | $3.76_{\pm 0.40}$ |
| | Post-hoc | 100 | $1.040_{\pm 0.019}$ | 0.606 | $0.435_{\pm 0.019}$ | 72 | $24.39_{\pm 0.43}$ | 19.52 | $4.87_{\pm 0.43}$ |
| | $\Delta$ | ↑ $67_{\pm 35}$ | ↑ $0.029_{\pm 0.023}$ | ↓ 0.063 | – | ↓ $19_{\pm 4}$ | ↑ $1.27_{\pm 0.26}$ | ↑ 0.16 | – |
| C-100-N-C Noisy | Naive | $8_{\pm 2}$ | $1.431_{\pm 0.008}$ | 1.234 | $0.198_{\pm 0.008}$ | $40_{\pm 41}$ | $41.42_{\pm 0.45}$ | 34.42 | $7.00_{\pm 0.45}$ |
| | Post-hoc | 32 | $1.744_{\pm 0.049}$ | 1.150 | $0.594_{\pm 0.049}$ | 38 | $45.45_{\pm 0.98}$ | 33.54 | $11.91_{\pm 0.98}$ |
| | $\Delta$ | ↑ $24_{\pm 2}$ | ↑ $0.313_{\pm 0.048}$ | ↓ 0.084 | – | ↓ $2_{\pm 41}$ | ↑ $4.03_{\pm 1.13}$ | ↓ 0.88 | – |
| C-100-N-F Clean | Naive | $93_{\pm 5}$ | $1.508_{\pm 0.017}$ | 1.065 | $0.443_{\pm 0.017}$ | $88_{\pm 5}$ | $33.83_{\pm 0.37}$ | 29.90 | $3.93_{\pm 0.37}$ |
| | Post-hoc | 75 | $1.567_{\pm 0.019}$ | 1.063 | $0.504_{\pm 0.019}$ | 95 | $33.86_{\pm 0.53}$ | 29.94 | $3.92_{\pm 0.53}$ |
| | $\Delta$ | ↓ $18_{\pm 5}$ | ↑ $0.059_{\pm 0.014}$ | ↓ 0.002 | – | ↑ $7_{\pm 5}$ | ↑ $0.03_{\pm 0.31}$ | ↑ 0.04 | – |
| C-100-N-F Noisy | Naive | $7_{\pm 2}$ | $2.416_{\pm 0.022}$ | 2.129 | $0.287_{\pm 0.022}$ | $91_{\pm 7}$ | $58.68_{\pm 0.49}$ | 51.34 | $7.34_{\pm 0.49}$ |
| | Post-hoc | 27 | $3.015_{\pm 0.079}$ | 1.994 | $1.021_{\pm 0.079}$ | 32 | $63.53_{\pm 0.55}$ | 50.26 | $13.27_{\pm 0.55}$ |
| | $\Delta$ | ↑ $20_{\pm 2}$ | ↑ $0.598_{\pm 0.075}$ | ↓ 0.135 | – | ↓ $59_{\pm 7}$ | ↑ $4.85_{\pm 0.60}$ | ↓ 1.08 | – |

# D   Post-Hoc Selection Results for Remaining Datasets

Table 5 compares naive and post-hoc selection for datasets not covered in the main paper. Post-hoc selection is mostly better than naive selection, although with varying margins. Post-hoc selection is sometimes worse, but only marginally[8].

# E   Detailed Results

Tables 6, 7, and 8 provide detailed results for CIFAR-N, LLM instruction tuning, and other datasets respectively.

# F   Optimal Checkpointing for Small Number of Epochs

Throughout the main paper, we use a checkpoint interval of 1 epoch. For small-epoch settings, such as LLM pre-training or fine-tuning, it might be better to checkpoint more frequently, at fractional epochs. In this section, we investigate the impact of checkpoint interval on the best MMLU score, and the epoch at which it is achieved, for the LLM instruction tuning setup of § 7.1.

---

[8]This may be attributed to (1) picking the same epoch for all runs in post-hoc selection, and (2) generalization error between validation and test sets for the selected epoch.

Table 7: Detailed results for LLM instruction tuning. Better values are in bold. Since Base and Gain columns involve 8 individual runs, we report mean$_{\pm\text{std. dev.}}$ of the metric.

| Transform → | None | SWA+TS | | SWA+Ens+TS | |
|---|---|---|---|---|---|
| Metric ↓ | | Naive | Ours | Naive | Ours |
| Perplexity | 3.756 | 3.471 | **3.461** | 3.245 | **3.142** |
| Error | 32.84 | 30.81 | **29.68** | 30.16 | **28.93** |
| MMLU | 46.64 | 46.78 | **47.03** | 47.23 | **47.54** |

Table 8: Detailed results for other datasets. See Table 6 caption for a description.

| Objective → | | Test Loss | | | | Test Error (%) | | | |
|---|---|---|---|---|---|---|---|---|---|
| Dataset ↓ | Select ↓ | Epochs | Base | Final | Gain | Epochs | Base | Final | Gain |
| FMoW (ID) | Naive | $2_{\pm0}$ | $1.583_{\pm0.014}$ | 1.494 | $0.089_{\pm0.014}$ | $15_{\pm19}$ | $43.20_{\pm0.46}$ | 37.95 | $5.24_{\pm0.46}$ |
| | Post-hoc | 50 | $2.831_{\pm0.053}$ | 1.305 | $1.526_{\pm0.053}$ | 48 | $43.18_{\pm0.55}$ | 34.93 | $8.24_{\pm0.55}$ |
| | Δ | ↑ $48_{\pm0}$ | ↑ $1.248_{\pm0.062}$ | ↓ 0.189 | – | ↑ $33_{\pm19}$ | ↓ $0.02_{\pm0.80}$ | ↓ 3.02 | – |
| FMoW (OOD) | Naive | $2_{\pm0}$ | $1.831_{\pm0.018}$ | 1.700 | $0.131_{\pm0.018}$ | $3_{\pm1}$ | $49.32_{\pm0.38}$ | 46.74 | $2.58_{\pm0.38}$ |
| | Post-hoc | 50 | $3.399_{\pm0.050}$ | 1.571 | $1.828_{\pm0.050}$ | 50 | $50.08_{\pm0.38}$ | 41.56 | $8.52_{\pm0.38}$ |
| | Δ | ↑ $48_{\pm0}$ | ↑ $1.567_{\pm0.054}$ | ↓ 0.129 | – | ↑ $47_{\pm1}$ | ↑ $0.75_{\pm0.66}$ | ↓ 5.19 | – |
| Yelp | Naive | $2_{\pm1}$ | $0.908_{\pm0.008}$ | 0.841 | $0.067_{\pm0.008}$ | $9_{\pm8}$ | $39.41_{\pm0.76}$ | 36.18 | $3.23_{\pm0.76}$ |
| | Post-hoc | 3 | $0.990_{\pm0.044}$ | 0.824 | $0.166_{\pm0.044}$ | 3 | $40.28_{\pm1.29}$ | 36.14 | $4.14_{\pm1.29}$ |
| | Δ | ↑ $1_{\pm1}$ | ↑ $0.082_{\pm0.040}$ | ↓ 0.017 | – | ↓ $6_{\pm8}$ | ↑ $0.87_{\pm1.16}$ | ↓ 0.04 | – |
| Income | Naive | $5_{\pm1}$ | $0.393_{\pm0.001}$ | 0.388 | $0.005_{\pm0.001}$ | $7_{\pm2}$ | $17.84_{\pm0.15}$ | 17.62 | $0.22_{\pm0.15}$ |
| | Post-hoc | 11 | $0.421_{\pm0.007}$ | 0.385 | $0.036_{\pm0.007}$ | 19 | $19.21_{\pm0.14}$ | 17.40 | $1.81_{\pm0.14}$ |
| | Δ | ↑ $6_{\pm1}$ | ↑ $0.028_{\pm0.006}$ | ↓ 0.003 | – | ↑ $12_{\pm2}$ | ↑ $1.37_{\pm0.22}$ | ↓ 0.22 | – |
| Public Coverage | Naive | $10_{\pm2}$ | $0.544_{\pm0.001}$ | 0.538 | $0.006_{\pm0.001}$ | $12_{\pm3}$ | $27.52_{\pm0.24}$ | 27.25 | $0.28_{\pm0.24}$ |
| | Post-hoc | 18 | $0.554_{\pm0.002}$ | 0.538 | $0.016_{\pm0.002}$ | 22 | $27.96_{\pm0.21}$ | 27.02 | $0.94_{\pm0.21}$ |
| | Δ | ↑ $8_{\pm2}$ | ↑ $0.010_{\pm0.002}$ | ↓ 0.000 | – | ↑ $10_{\pm3}$ | ↑ $0.44_{\pm0.25}$ | ↓ 0.22 | – |
| Mobility | Naive | $6_{\pm2}$ | $0.474_{\pm0.002}$ | 0.471 | $0.003_{\pm0.002}$ | $13_{\pm5}$ | $21.43_{\pm0.18}$ | 21.17 | $0.26_{\pm0.18}$ |
| | Post-hoc | 14 | $0.476_{\pm0.003}$ | 0.468 | $0.008_{\pm0.003}$ | 11 | $21.40_{\pm0.17}$ | 21.24 | $0.16_{\pm0.17}$ |
| | Δ | ↑ $8_{\pm2}$ | ↑ $0.002_{\pm0.003}$ | ↓ 0.003 | – | ↓ $2_{\pm5}$ | ↓ $0.03_{\pm0.22}$ | ↑ 0.07 | – |
| Employment | Naive | $8_{\pm1}$ | $0.380_{\pm0.000}$ | 0.378 | $0.003_{\pm0.000}$ | $14_{\pm4}$ | $17.94_{\pm0.08}$ | 17.72 | $0.22_{\pm0.08}$ |
| | Post-hoc | 15 | $0.383_{\pm0.001}$ | 0.377 | $0.006_{\pm0.001}$ | 30 | $18.27_{\pm0.12}$ | 17.83 | $0.43_{\pm0.12}$ |
| | Δ | ↑ $7_{\pm1}$ | ↑ $0.003_{\pm0.001}$ | ↓ 0.000 | – | ↑ $16_{\pm4}$ | ↑ $0.33_{\pm0.16}$ | ↑ 0.11 | – |
| Travel Time | Naive | $6_{\pm2}$ | $0.597_{\pm0.002}$ | 0.596 | $0.001_{\pm0.002}$ | $9_{\pm1}$ | $35.77_{\pm0.34}$ | 35.44 | $0.32_{\pm0.34}$ |
| | Post-hoc | 15 | $0.626_{\pm0.003}$ | 0.591 | $0.035_{\pm0.003}$ | 16 | $36.40_{\pm0.20}$ | 35.23 | $1.17_{\pm0.20}$ |
| | Δ | ↑ $8_{\pm2}$ | ↑ $0.029_{\pm0.003}$ | ↓ 0.005 | – | ↑ $7_{\pm1}$ | ↑ $0.64_{\pm0.42}$ | ↓ 0.21 | – |
| Collab | Naive | $52_{\pm18}$ | $0.492_{\pm0.044}$ | 0.439 | $0.053_{\pm0.044}$ | $75_{\pm28}$ | $20.65_{\pm1.06}$ | 20.40 | $0.25_{\pm1.06}$ |
| | Post-hoc | 163 | $1.075_{\pm0.122}$ | 0.404 | $0.671_{\pm0.122}$ | 152 | $20.95_{\pm1.26}$ | 18.80 | $2.15_{\pm1.26}$ |
| | Δ | ↑ $111_{\pm18}$ | ↑ $0.583_{\pm0.146}$ | ↓ 0.035 | – | ↑ $77_{\pm28}$ | ↑ $0.30_{\pm1.31}$ | ↓ 1.60 | – |
| Reddit-5k | Naive | $15_{\pm4}$ | $1.154_{\pm0.022}$ | 1.101 | $0.053_{\pm0.022}$ | $13_{\pm5}$ | $47.42_{\pm0.64}$ | 47.09 | $0.33_{\pm0.64}$ |
| | Post-hoc | 44 | $1.448_{\pm0.058}$ | 1.085 | $0.362_{\pm0.058}$ | 45 | $50.75_{\pm1.83}$ | 45.49 | $5.26_{\pm1.83}$ |
| | Δ | ↑ $29_{\pm4}$ | ↑ $0.294_{\pm0.059}$ | ↓ 0.015 | – | ↑ $32_{\pm5}$ | ↑ $3.33_{\pm2.05}$ | ↓ 1.60 | – |
| Reddit-12k | Naive | $16_{\pm3}$ | $1.405_{\pm0.011}$ | 1.367 | $0.038_{\pm0.011}$ | $17_{\pm4}$ | $51.78_{\pm1.05}$ | 50.34 | $1.45_{\pm1.05}$ |
| | Post-hoc | 41 | $1.585_{\pm0.023}$ | 1.346 | $0.239_{\pm0.023}$ | 64 | $55.85_{\pm0.97}$ | 51.26 | $4.59_{\pm0.97}$ |
| | Δ | ↑ $25_{\pm3}$ | ↑ $0.180_{\pm0.027}$ | ↓ 0.021 | – | ↑ $47_{\pm4}$ | ↑ $4.07_{\pm1.59}$ | ↑ 0.92 | – |

Figs. 14a and 14b show the results. We find that a checkpointing interval of 0.7 epochs gives the best results, with higher and lower intervals performing slightly worse. This makes sense—higher intervals include too few checkpoints for SWA, lower ones include too many weaker checkpoints from earlier in training.

Also, we find that the optimal epoch is shifted further at smaller checkpointing intervals (by about 2 epochs when the checkpointing interval is 0.1 epochs), showing that **post-hoc reversal is even more important** in this setting. This is likely because with more checkpoints being averaged, even more overfitted checkpoints can be accomodated while still increasing the overall performance.

## G  Visualizing Post-Hoc Reversal on a Synthetic Dataset

Here, we replicate post-hoc reversal on a synthetic dataset with 2 input features, with the aim of visualizing learnt decision surfaces to solidify our intuitions.

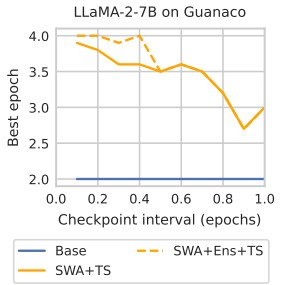
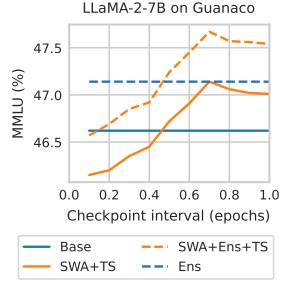
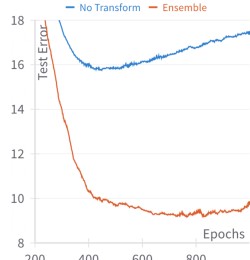

(a) *Best epoch vs checkpoint freq.*     (b) *MMLU vs checkpoint freq.*

Figure 14: Best MMLU and epoch at which it is achieved vs checkpointing interval, for the LLM instruction tuning setup of § 7.1. Checkpointing every 0.7 epochs gives the best results. Best epoch is shifted further at smaller checkpointing intervals, i.e. post-hoc reversal is more prominent in this setting.

Figure 15: The synthetic dataset setup in § G exhibits post-hoc reversal between epochs 440 and 1000.

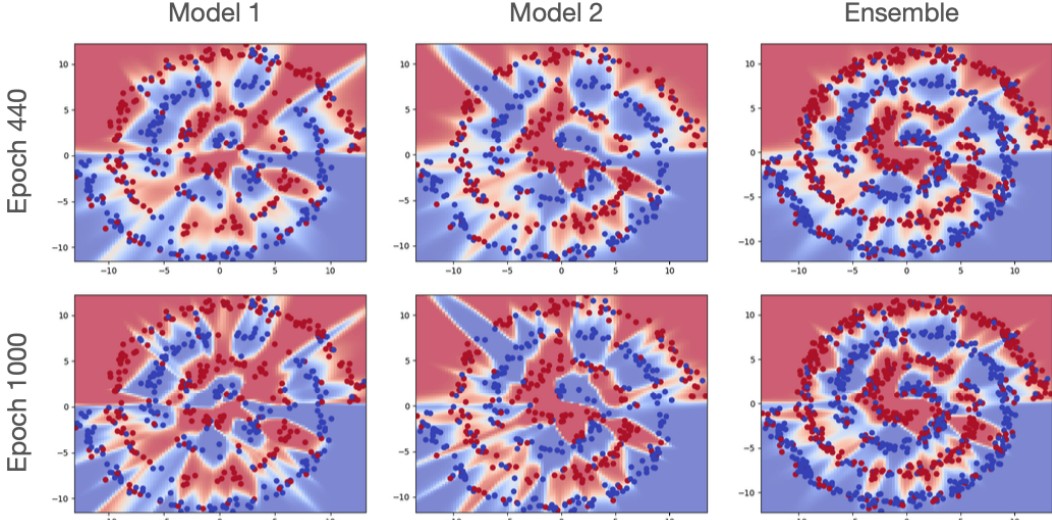

Figure 16: Decision surfaces of 2 models and the ensemble (of 16 models) on a synthetic 2D dataset of spirals, at epochs 440 and 1000, between which post-hoc reversal occurs (Fig. 15).

We train 4-layer MLPs with 512 ReLU units per hidden layer on a 2-class spirals dataset of 1000 training examples, with 20% of the labels flipped at random. We train 16 MLPs and track the mean test error across epochs, as well as the test error of the ensemble (Fig. 15).

As per [3, 16] ensembling and SWA help when the data has a "multi-view" structure, or equivalently, the loss landscape has multiple modes. This is hard to achieve for 2D datasets, so instead we simulate the effect by training each MLP on a random 50% subsample of the training data.

Fig. 16 shows decision surfaces at epochs 440 and 1000 for 2 MLPs and the ensemble. Decision boundaries are spiky around noisy examples and smoother around clean ones. While the generalizable parts of the spiral are retained in the ensemble, the effects of noisy examples are diminished. Between epochs 440 and 1000, individual models spike around noisy examples more prominently than they learn new parts of the spiral, but the ensemble surface is relatively unchanged, except for small improvements to learning the spiral.

This reinforces our intuitions from § 5 that mislabeled examples have a more unstable influence on the decision boundary, and post-hoc transforms exploit this to reduce their impact, while amplifying generalizable patterns learnt from clean examples.

Table 9: Naive vs post-hoc (ours) selection for CIFAR-N trained with **cross-entropy (CE)** loss. Better values are in bold.

| Metric → | Test Loss | | | | | Test Error (%) | | | | |
|---|---|---|---|---|---|---|---|---|---|---|
| Transform → | None | SWA+TS | | SWA+Ens+TS | | None | SWA+TS | | SWA+Ens+TS | |
| Dataset ↓ | | Naive | Ours | Naive | Ours | | Naive | Ours | Naive | Ours |
| C-10-N Clean | 0.435 | **0.269** | 0.270 | 0.234 | **0.233** | 9.75 | **9.09** | 9.10 | 8.30 | **8.24** |
| C-10-N Aggre | 0.722 | 0.663 | **0.585** | 0.608 | **0.543** | 19.20 | 17.08 | **16.95** | 15.88 | **15.74** |
| C-10-N Rand1 | 1.009 | 0.968 | **0.907** | 0.916 | **0.859** | 28.63 | 27.13 | **24.84** | 24.80 | **23.50** |
| C-10-N Rand2 | 1.040 | 0.983 | **0.935** | 0.931 | **0.888** | 29.91 | 27.60 | **25.69** | 25.44 | **24.12** |
| C-10-N Rand3 | 1.005 | 0.963 | **0.911** | 0.910 | **0.864** | 28.96 | 26.91 | **25.09** | 24.86 | **23.44** |
| C-10-N Worst | 1.511 | 1.483 | **1.443** | 1.437 | **1.399** | 46.84 | 46.12 | **44.14** | 44.30 | **42.88** |
| Clean | 1.011 | 0.786 | **0.686** | 0.669 | **0.606** | 23.12 | **21.30** | 21.38 | **19.36** | 19.52 |
| Noisy | 1.431 | 1.330 | **1.235** | 1.234 | **1.150** | 41.42 | 38.08 | **35.87** | 34.42 | **33.54** |
| C-100-N Clean | 1.508 | 1.215 | **1.205** | 1.065 | **1.063** | 33.83 | **32.67** | 32.69 | **29.90** | 29.94 |
| C-100-N Noisy | 2.416 | 2.289 | **2.136** | 2.129 | **1.994** | 58.68 | 54.94 | **53.18** | 51.34 | **50.26** |

Table 10: Naive vs post-hoc (ours) selection for CIFAR-N trained with **SOP** loss. Better values are in bold.

| Metric → | Test Loss | | | | | Test Error (%) | | | | |
|---|---|---|---|---|---|---|---|---|---|---|
| Transform → | None | SWA+TS | | SWA+Ens+TS | | None | SWA+TS | | SWA+Ens+TS | |
| Dataset ↓ | | Naive | Ours | Naive | Ours | | Naive | Ours | Naive | Ours |
| C-10-N Clean | 0.425 | 0.270 | **0.269** | 0.236 | **0.235** | 9.65 | 8.82 | **8.81** | **7.96** | 8.00 |
| C-10-N Aggre | 0.728 | 0.693 | **0.573** | 0.634 | **0.541** | 18.03 | **16.55** | 16.56 | 15.58 | **15.56** |
| C-10-N Rand1 | 1.025 | 0.980 | **0.888** | 0.925 | **0.851** | 26.91 | 24.53 | **24.50** | 23.20 | **23.14** |
| C-10-N Rand2 | 1.045 | 1.015 | **0.920** | 0.957 | **0.883** | 27.39 | 25.34 | **25.25** | **24.12** | 24.16 |
| C-10-N Rand3 | 1.016 | 0.975 | **0.889** | 0.921 | **0.851** | 26.66 | **24.23** | 24.23 | 23.02 | **22.96** |
| C-10-N Worst | 1.514 | 1.492 | **1.451** | 1.447 | **1.413** | 46.78 | 46.26 | **44.29** | 44.50 | **42.78** |
| Clean | 1.018 | 0.742 | **0.686** | 0.623 | **0.608** | 23.07 | **21.43** | 21.47 | **19.18** | 19.78 |
| Noisy | 1.427 | 1.347 | **1.229** | 1.247 | **1.145** | 41.39 | 38.01 | **35.85** | 34.32 | **33.94** |
| C-100-N Clean | 1.513 | 1.213 | **1.203** | 1.063 | **1.061** | 33.79 | **32.66** | 32.68 | **29.46** | 29.56 |
| C-100-N Noisy | 2.415 | 2.268 | **2.137** | 2.118 | **1.997** | 58.34 | 54.76 | **53.48** | 51.06 | **50.54** |

# H  Noise-Aware Training

While our experiments in the main paper use the standard cross-entropy (CE) loss, here we consider two leading training objectives from the label noise literature: (1) SOP [45] and (2) ELR [42]. Tables 9, 10 and 11 compare naive and post-hoc selection strategies for CIFAR-N datasets under CE, SOP and ELR losses respectively. Here again we find that post-hoc selection is superior to naive selection in general. We also note that the differences between CE, SOP and ELR are minimal. This is likely because we use i.i.d. (and therefore noisy) validation and test sets, unlike the original papers which use clean validation and test sets.

Table 11: Naive vs post-hoc (ours) selection for CIFAR-N trained with **ELR** loss. Better values are in bold.

| Metric → | Test Loss | | | | | Test Error (%) | | | | |
|---|---|---|---|---|---|---|---|---|---|---|
| Transform → | None | SWA+TS | | SWA+Ens+TS | | None | SWA+TS | | SWA+Ens+TS | |
| Dataset ↓ | | Naive | Ours | Naive | Ours | | Naive | Ours | Naive | Ours |
| C-10-N Clean | 0.421 | 0.271 | **0.269** | 0.233 | **0.232** | 9.53 | **8.92** | 9.01 | 7.98 | **7.92** |
| C-10-N Aggre | 0.730 | 0.659 | **0.584** | 0.606 | **0.541** | 19.02 | 16.86 | **16.80** | **15.34** | 15.52 |
| C-10-N Rand1 | 1.019 | 0.975 | **0.911** | 0.921 | **0.864** | 29.42 | 26.68 | **24.86** | 24.30 | **23.56** |
| C-10-N Rand2 | 1.042 | 0.994 | **0.939** | 0.941 | **0.893** | 29.79 | 27.98 | **25.74** | 26.12 | **24.50** |
| C-10-N Rand3 | 1.004 | 0.964 | **0.913** | 0.912 | **0.866** | 28.80 | 26.68 | **24.84** | 24.52 | **23.32** |
| C-10-N Worst | 1.508 | 1.492 | **1.443** | 1.444 | **1.397** | 46.94 | 46.27 | **44.03** | 44.64 | **42.48** |
| Clean | 1.030 | 0.760 | **0.686** | 0.644 | **0.605** | 23.07 | **21.27** | 21.40 | 19.28 | **19.24** |
| Noisy | 1.415 | 1.317 | **1.236** | 1.228 | **1.152** | 41.55 | 38.40 | **35.74** | 34.72 | **33.60** |
| C-100-N Clean | 1.518 | 1.223 | **1.210** | 1.070 | **1.068** | 34.05 | **32.92** | 32.97 | 29.68 | **29.66** |
| C-100-N Noisy | 2.432 | 2.287 | **2.140** | 2.130 | **1.997** | 58.85 | 54.84 | **53.24** | 50.86 | **50.50** |

# I  Limitations

We find post-hoc reversal to be an important phenomenon when the base curve exhibits performance degradation due to overfitting. However, under some scenarios, the base curve shows a monotonic improvement in performance with additional training (or increasing model size). Examples include: (1) the data has low noise, (2) the training is heavily regularized, and (3) there is an abundance of data, so that a single data point is not repeated enough to cause overfitting. In such cases, post-hoc selection outcomes are similar to naive selection. Since our suggested approach only ensembles models trained for the same number of epochs during post-hoc selection, it does not subsume the naive selection search space, leading to marginally worse performance sometimes, although this can be easily overcome in practice.

