# OpenReview forum: "Post-Hoc Reversal: Are We Selecting Models Prematurely?"
_NeurIPS.cc/2024/Conference — NeurIPS 2024 poster_

### Official Review · Reviewer_GiJ5 · 2024-07-08

**Soundness:** 3
**Presentation:** 3
**Contribution:** 3
**Rating:** 7
**Confidence:** 3

**Summary:**

This paper shows empirical evidence that common approaches to model selection can be improved upon by less greedy alternatives. In particular, authors highlighted what they referred to as post-hoc reversal, when post-training transformations reverse the trends observed in independent training runs. For instance, in certain overfitting situations, transformations such as temperature scaling, weight averaging, and ensembling, reversed growing test error trends and closed the generalization gap to a greater extent than common model selection approaches. This observation then yielded practical recommendations of incorporating post-training transforms into the model selection pipeline, which resulted in improved performance across a number of settings, as well as on smoother error curves that render model selection less noisy. Authors attributed post-hoc reversal to the high variance involved in the training of neural networks, which can be smoothed out by some of the studied transformations. The paper lays out clear practical recommendations as a result of the reported empirical observations, which I believe would be highly relevant to the community.

**Strengths:**

- The paper is well written and easy to follow, and tackles a highly relevant problem in practice.

- The evaluation is broad and covers a number of model classes and sizes, and data modalities.

- Conclusions lay out very clear and easy to implement recommendations to improve model selection.

**Weaknesses:**

- This requires no action from the authors but just for the record, I have mixed feelings about the presentation choices regarding the use of notation. The text is notation heavy, which makes it precise, but at the cost of readability. The discussion seems simple enough to enable the use of almost plain text only. But I reiterate that I consider this to be mostly matter of writing style, and I expect no effort from authors addressing this comment.

- The bulk of the empirical assessment focuses on a somewhat small scale setting involving variations of the Cifar-10 dataset and relatively small neural networks. While this is obviously due to the high cost involved in replicating these experiments in larger scale settings, it limits the strength of the presented empirical evidence. However, section 6 adds results for other settings including fine-tuning of very large models, which address this concern to an extent.

- The experiments focus on multi-epoch training settings, and it's a bit unclear how those results transfer to now common pre-training situations where very few epochs are used. While one can replace epoch-end checkpoints by checkpoints obtained every k steps, it's unclear how the choice of k affects results, for instance.

- One component that seems to have been left out of the analysis is robustness. Does post-hoc reversal still happens in situations where the test set is somehow shifted relative to the validation set used for post-hoc selection? It might be the case that there's not so pronounced of a gap between post-hoc and greedy/naive selection in such a scenario.

**Questions:**

- Would it be possible to replicate a subset of the results reported in section 4 and 5 for a larger scale dataset (e.g., ImageNet)?

- Does post-hoc reversal still happen in situations where the test set is somehow shifted relative to the validation set used for post-hoc selection?

- How does one leverage post-hoc selection in a non-multi-epoch training situation?

**Limitations:**

Please refer to the "Weaknesses" section.

---

> ### Author Rebuttal · Authors · 2024-08-06
>
> Dear reviewer GiJ5,
>
> Thank you for your review
> and thoughtful suggestions about writing.
> We will take them into account
> while revising the manuscript.
> We address your concerns regarding experiments below.
>
> > Would it be possible to replicate a subset of the results reported in section 4 and 5 for a larger scale dataset (e.g., ImageNet)?
>
> Due to limited time in the rebuttal period,
> we are unable to report results on ImageNet
> at this time.
> However, we appreciate this suggestion
> and will look into incorporating larger scale datasets
> in the final version of this work.
>
> > Does post-hoc reversal still happen in situations where the test set is somehow shifted relative to the validation set used for post-hoc selection?
>
> Yes.
> We report an experimental analysis
> on the FMoW dataset below.
>
> ## Experiment 1:
> We used the distribution-shifted
> val and test sets provided by WILDS [1].
> Here, the val set is shifted w.r.t. the train set
> and the test set is shifted w.r.t. both the train and val sets.
> The rest of the setup is same
> as for Table 1 in the paper.
> For convenient comparison,
> we also reproduce the
> in-distribution FMoW numbers
> from the paper.
> The in-distribution FMoW dataset is denoted with ID,
> while the distribution-shifted (out-of-distribution)
> version is denoted with OOD.
> The 4 tables below
> show the results for
> the test loss and test error metrics
> and for the SWA+TS and SWA+Ens+TS transforms.
> _Base_ column has numbers without any post-hoc transform,
> while _Naive_ and _Ours_ represent
> the application of post-hoc transforms
> with naive and post-hoc selection strategies respectively.
>
>
> __Test Loss, SWA+TS:__
> |     | Base  | Naive | Ours      | Diff  |
> |-----|-------|-------|-----------|-------|
> | ID  | 1.583 | 1.627 | **1.554** | 0.073 |
> | OOD | 1.831 | 1.840 | **1.788** | 0.052 |
>
> __Test Loss, SWA+Ens+TS:__
> |     | Base  | Naive | Ours      | Diff  |
> |-----|-------|-------|-----------|-------|
> | ID  | 1.583 | 1.494 | **1.305** | 0.189 |
> | OOD | 1.831 | 1.700 | **1.571** | 0.129 |
>
> __Test Error (%), SWA+TS:__
> |     | Base  | Naive | Ours      | Diff  |
> |-----|-------|-------|-----------|-------|
> | ID  | 43.20 | 42.69 | **39.92** | 2.77  |
> | OOD | 49.32 | 49.70 | **46.75** | 2.95  |
>
> __Test Error (%), SWA+Ens+TS:__
> |     | Base  | Naive | Ours      | Diff  |
> |-----|-------|-------|-----------|-------|
> | ID  | 43.20 | 37.95 | **34.93** | 3.02  |
> | OOD | 49.32 | 46.74 | **41.56** | 5.18  |
>
>
> We observe that post-hoc selection is about as effective
> in the OOD case as in the ID case.
> Interestingly, the improvement
> for OOD as compared to ID
> is slightly lower for test loss
> but higher for test error.
>
> Thank you for suggesting this experiment.
> We will add it to the paper.
>
>
> > The experiments focus on multi-epoch training settings, and it's a bit unclear how those results transfer to now common pre-training situations where very few epochs are used. While one can replace epoch-end checkpoints by checkpoints obtained every k steps, it's unclear how the choice of k affects results, for instance.
>
> Thank you for suggesting this analysis;
> we believe it would be a valuable addition to the paper.
> We have conducted this experiment
> on our LLM instruction tuning setup from Section 6.1,
> where best results are obtained within 3-4 epochs.
>
> ## Experiment 2:
> We vary the checkpointing interval
> as a fraction of 1 epoch,
> and record the best test accuracy,
> as well as the epoch at which it is obtained
> for both SWA+TS and SWA+Ens+TS.
>
> Figs. 4 (a) and (b) in the PDF attached to the global rebuttal
> shows the results.
> We find that a checkpointing interval of 0.7 epochs gives the best results,
> with higher and lower intervals performing slightly worse.
> This makes sense:
> higher intervals include too few checkpoints for SWA,
> lower ones include too many weaker checkpoints from earlier in training.
>
> Also, we find that the optimal epoch is shifted
> further at smaller checkpointing intervals
> (by about 2 epochs when the checkpointing interval is 0.1 epochs),
> showing that post-hoc reversal is even more important
> in this setting.
> This is likely because with more checkpoints being averaged,
> even more overfitted checkpoints can be accomodated
> while still increasing the overall performance.
>
>
> Please let us know if any of your concerns are still unaddressed.
>
> Thanks,
>
> Authors
>
> [1] WILDS: A Benchmark of in-the-Wild Distribution Shifts

---

> > ### Comment · Reviewer_GiJ5 · 2024-08-13
> >
> > Thank you for the clarifications and extra results.

---

### Official Review · Reviewer_NSoG · 2024-07-11

**Soundness:** 3
**Presentation:** 3
**Contribution:** 2
**Rating:** 5
**Confidence:** 3

**Summary:**

The authors investigate applying three post hoc transforms, namely temperature scaling (TS), ensembling, and stochastic weight averaging (SWA), to trained models after training, separately and jointly. They oppose their approach to the "naive approach" and provide empirical observations of a phenomenon they refer to as "post hoc reversal."  This phenomenon corresponds to the performance trends of models that change after these transformations. In particular, they focus on the observation in noisy settings of the following observations.

1- Epoch-wise post hoc reversal. The authors show that post hoc transforms reduce overfitting, double descent, and loss-error mismatch. In particular, SWA and ensembling reduce overfitting and flatten double descent peaks.

2- Model-wise post hoc reversal. The model's size influences the improvement in the performance of the post hoc transforms, i.e., larger models tend to perform better under post hoc transforms.

3- Hyperparameter-Wise Post Hoc Reversal. It appears that constant learning rates improve performance better than decaying learning ones and the optimal number of epoch shifts.

The experiments are broad, and metrics show good non-trivial improvement based on the method.

**Strengths:**

1 - The paper provides an extensive empirical study across various domains (vision, language, tabular, graph), aiming to demonstrate the generality of their approach, i.e., post hoc transforms.

2- The authors use a variety of performance metrics (error rates, loss, MMLU accuracy) to validate the effectiveness of post hoc transforms, leading to a robust evaluation.

3- The authors identify a potentially impactful post hoc reversal phenomenon that could challenge commonly adopted practices.

4- The approach seems to lead to consistent performance gains across different datasets and settings. The focus is on noisy environments, which can be particularly relevant for less curated datasets, which might be what the field needs.

5- The authors suggest that their approach could lead to guiding principles for model development, including early stopping and checkpoint selection.

**Weaknesses:**

The main weakness of the approach, which is recurrent in most current empirical LLM observations, is that understanding the source of the improvement needs to be clarified.

In fact, it is not clear why the benefits of post hoc transforms are more pronounced in high-noise settings.

Making assumptions to verify potentially synthetic and controlled datasets could be helpful. As of one, this corresponds to an observation.

**Questions:**

If the author can provide a good answer to the question above that would address my concerns.

**Limitations:**

Yes

---

> ### Author Rebuttal · Authors · 2024-08-06
>
> Dear reviewer NSoG,
>
> Thank you for your review.
> We provide a detailed reponse
> regarding the intuitions and explanations for post-hoc reversal (PHR)
> in the global rebuttal, along with experimental analyses on CIFAR-10N
> and a synthetic dataset to back our claims.
> Below we highlight how the global rebuttal answers your questions.
>
> > the source of the improvement needs to be clarified.
>
> Even when performance degrades due to overfitting,
> the model continues to learn generalizable patterns
> from the clean training points,
> but this is outweighed by the spurious patterns
> learnt from the noisy training points.
> Under post-hoc transforms,
> the generalizable patterns reinforce each other,
> whereas the spurious patterns cancel out.
> This is responsible for the performance improvement
> seen under PHR.
>
> We expand on this intuition in the [Intuition 3] section of the global rebuttal,
> and further demonstrate it for CIFAR-10N under Experiment 2,
> and for a synthetic dataset under Experiment 3.
>
>
> > it is not clear why the benefits of post hoc transforms are more pronounced in high-noise settings.
>
> In high-noise settings, noisy training points have
> a greater adverse effect on the learnt model.
> Post-hoc transforms subdue this to a large extent.
> This produces a more pronounced benefit.
> In contrast, the base models themselves perform well in the low-noise setting,
> leaving less room for improvement by post-hoc transforms.
>
> We give a more detailed explanation in the global rebuttal,
> separately discussing the mechanisms by which temperature scaling
> and ensembling/SWA operate in the presence of noise
> ([Intuition 2] and [Intuition 3] sections respectively),
> thereby elucidating their increased efficacy with higher noise levels.
>
>
> > Making assumptions to verify potentially synthetic and controlled datasets could be helpful.
>
> Thank you for suggesting this.
> In the global rebuttal,
> we replicate post-hoc reversal
> on a synthetic controlled-noise dataset with 2 input features,
> and visualize the learnt decision surfaces to
> verify our proposed intuitions
> (Experiment 3).
> We plan to further improve this analysis
> and incorporate it in our paper,
> along with the intuitions/explanations above.
>
> Please let us know if any of your concerns are still unaddressed.
>
> Thanks,
>
> Authors

---

### Official Review · Reviewer_vEtm · 2024-07-13

**Soundness:** 2
**Presentation:** 3
**Contribution:** 2
**Rating:** 5
**Confidence:** 4

**Summary:**

The paper discusses the phenomenon of *post-hoc reversal*, where the performance trend is reversed after applying post hoc transforms, namely, temperature scaling (TS), stochastic weight averaging (SWA), and ensembling (Ens).

The paper conducts an empirical study to observe the phenomenon across different epochs, model sizes, and hyperparameters.

They propose a *post-hoc selection* strategy where the optimal epoch/model size/hyperparameters are selected considering the performance after applying post-hoc transforms.

The paper focuses on the noisy data setting and shows experimental gains in that setting for some datasets.

Experiments have been conducted for the FMoW dataset, CIFAR-10-N, and CIFAR-100-N, as well as some additional experiments on text, tabular, and graph datasets.

**Strengths:**

1. The paper demonstrates the phenomenon of post-hoc reversal.
2. They conduct experiments on diverse domains.
3. Performance gains are achieved with the proposed strategy in some of the noisy data settings.

**Weaknesses:**

1. The paper proposes *post-hoc selection* strategy, simply selecting the optimal epoch (or model size, etc.) after applying already existing (and widely used) post-hoc transforms (SWA, En, TS). They do not propose any *novel* mechanism to tackle *post-hoc reversal*.
2. It is a findings paper; the study is empirical and lacks theoretical insights. Given the empirical nature of the study, the performance gains in some datasets are marginal (e.g., C-100-N Noisy test error, Table 1).
3. For the datasets in Table 5, the post-hoc selection strategy does not consistently provide performance gains, i.e., it performs worse in some cases.

**Questions:**

**Questions**
1. Why is post-hoc reversal prominent in the noisy data setting? Can the authors provide intuition or reasoning (other than empirical observations)?
2. Can the authors provide intuitions to (at least) some of the observations? For example, why do SWA and Ens handle double descent, but TS causes it?
3. In Figure 7, what is the SWA ensemble? It has not been discussed in Section 5.
4. For the datasets Yelp, Income, and Reddit-12K, the post-hoc reversal is observed in Figure 7, but the post-hoc selection strategy either shows worse performance or marginally better. Does this indicate that the post-hoc selection strategy is ineffective in these settings? Can the authors provide any reasoning/explanations?

**Suggestions**
* The paper title should include "noisy data setting", as the paper focuses on the noisy data setting.
* Include intuitions/reasonings/explanations (wherever possible) for the observations in the experimental results.

**Limitations:**

Yes.

---

> ### Author Rebuttal · Authors · 2024-08-06
>
> Dear reviewer vEtm,
>
> Thank you for your review, including suggestions to improve the paper.
> We address your concerns below:
>
> > They do not propose any _novel_ mechanism to tackle _post-hoc reversal_.
>
> First, we would like to clarify that
> post-hoc reversal (PHR) is not a problem.
> On the contrary,
> PHR usually provides
> an opportunity to improve performance.
> By post-hoc selection (PHS), we show that
> a simple change to existing methodology
> is sufficient to reap benefits.
> We leave smarter checkpoint selection to future work,
> as it is beyond the scope of this work,
> whose primary aim is to demonstrate and characterize PHR.
>
>
> > Given the empirical nature of the study, the performance gains in some datasets are marginal (e.g., C-100-N Noisy test error, Table 1).
>
> > For the datasets in Table 5, the post-hoc selection strategy does not consistently provide performance gains, i.e., it performs worse in some cases.
>
> When solely considering the test error metric, this is indeed true.
> Please note that test error is capped by the Bayes error and the model's strength.
> For example, C-100-N Noisy has \~40% noise and ResNet18 gets \~10% test error with clean data.
> Given this lower bound of \~50%, the achieved error of 50.26% could still be considered impressive.
>
> In fact, test error is not the best metric for datasets with high Bayes error.
> Hence our equal focus on the test loss metric, which evaluates predicted probabilities.
> We humbly point out that across _all_ our datasets and transforms, PHS outperforms naive selection, and in most cases quite significantly so.
>
> Even for test error, PHS is worse in only 5 of our 21 datasets, and except in one of these (Reddit-12k) it's less than 0.2 pts worse. Given that PHS is simple and cheap,
> we stand by our recommendation to employ it in practice.
>
>
> > Why is post-hoc reversal prominent in the noisy data setting? Can the authors provide intuition or reasoning (other than empirical observations)?
>
> Please see our global rebuttal
> where we provide detailed intuitions
> for PHR and highlight it's important connection to noise.
> Our explanations cover
> epoch-, model-, and hyperparamter-wise PHR;
> TS, Ens and SWA transforms;
> and imporant consequences such as to
> loss-error mismatch, catastrophic overfitting and double descent.
>
> We further validate our proposed intuitions
> with experimental analysis on CIFAR-10N,
> and visualization of the learnt decision surfaces
> on a synthetic dataset.
>
>
> > Can the authors provide intuitions to (at least) some of the observations? For example, why do SWA and Ens handle double descent, but TS causes it?
>
> We again refer you to our global rebuttal for
> comprehensive explanations of various observations.
> Here we restate our explanation for the particular observation mentioned,
> namely, why SWA and Ens handle double descent but TS causes it.
>
> In all our experiments,
> we do not observe double descent in the base test loss curves.
> Overfitting occurs too drastically in the test loss
> for the second descent to occur.
> This is because once a noisy training point is fit,
> the model can lower the loss simply by upscaling the logits.
> This leads to overconfident predictions
> and poor generalization loss around noisy training points.
> However, scaling the logits does not affect error.
> Indeed test error overfits less,
> sometimes even exhibiting a second descent,
> where the continued learning overpowers the overfitting.
> By rescaling the logits, temperature scaling
> does not so much cause double descent
> as it removes the loss-error mismatch.
> If the test error curve has a double descent,
> the post-TS test loss curve does too,
> simply because the post-TS loss tracks the error more closely.
>
> As mentioned earlier, we don't observe double descent in loss curves,
> so also no instances where TS removes double descent,
> as TS is unable to affect the test error metric.
>
> For intuitions on why Ens/SWA can suppress double descent,
> please refer to [Intuition 3] in the global rebuttal.
>
>
> > In Figure 7, what is the SWA ensemble? It has not been discussed in Section 5.
>
> SWA ensemble refers to the ensemble of SWA models,
> i.e., first apply SWA to checkpoints from the same training run,
> then ensemble the SWA models across different runs.
> Thank you for pointing this out.
> We will clarify in the manuscript.
>
>
> > For the datasets Yelp, Income, and Reddit-12K, the post-hoc reversal is observed in Figure 7, but the post-hoc selection strategy either shows worse performance or marginally better. Does this indicate that the post-hoc selection strategy is ineffective in these settings? Can the authors provide any reasoning/explanations?
>
> This is a fair observation.
> Looking carefully at Figure 7,
> one finds that while PHR occurs quite prominently,
> the optima of the post-hoc curves are only marginally better than
> the optima of the corresponding base curves.
> Another source of error is that while PHR curves
> are all drawn for the test set,
> for the PHS results,
> the epoch is selected based on the val set,
> and the reported numbers are evaluated on the test set.
>
> We believe that this does not diminish the usefulness
> of PHS, because it is never substantially worse
> than naive selection.
>
> Further, depending on the setting, one might improve results
> by ensembling more models.
> We ensemble 8 models throughout the paper for uniformity,
> but for small tabular datasets like Income,
> it is feasible to ensemble many more models.
>
> A final factor that may be playing a role here,
> is that post-hoc transforms under any selection strategy
> cannot be expected to give too much improvement,
> if the base models already achieves close to
> the Bayes error for the dataset,
> but this is hard to evaluate beforehand.
>
>
> Please let us know if any of your concerns are still unaddressed.
>
> Thanks,
>
> Authors

---

> > ### Comment · Reviewer_vEtm · 2024-08-12
> > **Acknowledgement to Author's Rebuttal**
> >
> > Thanks to the authors for the detailed response.
> >
> > Most of my queries have been addressed. I do not have any further questions at this point.

---

### Author Rebuttal · Authors · 2024-08-07

# Overview

We thank all the reviewers for their feedback
and helpful suggestions on improving the work.
We have added extensive explanations and intuitions,
backed by numerous additional experiments and analyses.
We summarize them below:
1. Epoch-, model-, and hyperparameter-wise post-hoc reversal (PHR)
can be viewed under the common lens of effective model complexity (EMC),
allowing us to focus on intuitions for epoch-wise reversal (Intuition 1).
2. Loss-error mismatch results from increasingly overconfident predictions
blowing up the test loss in the presence of noise, but not test error.
Temperature scaling fixes this by downscaling the logits,
resulting in PHR (Intuition 2).
3. Neural networks learn noisy points differently
than clean ones.
To wit,
predictions for noisy points fluctuate more during training
(see Expt. 1 on CIFAR-10N),
indicating unstable decision boundaries around them
(see Expt. 3 on synthetic dataset).
4. Building on the above,
ensembling and SWA act differently
on patterns learnt from clean and noisy points
(see Expt. 2 on CIFAR-10N),
reinforcing the former and
suppressing the latter
(see Expt. 3 on synthetic dataset).
Post-hoc reversal is observed when after transform
continued learning from clean points outshines
overfitting from noisy points (Intuition 3).
5. Post-hoc selection is effective even under distribution shift
(See Expt. 1 in response to reviewer GiJ5).
6. Post-hoc reversal is even more relevant for
checkpointing at fractional epochs in very-low-epoch settings.
(See Expt. 2 in response to reviewer GiJ5).

# [Intuition 1] Effective Model Complexity (EMC)

Epoch-, model-, and hyperparam-wise PHR can be unified via EMC,
introduced in [1] to unify epoch-, and model-wise double descent.
EMC measures memorization capacity,
and is important to us because
memorization of noisy point plays a key role in PHR.
EMC increases with epochs and model size,
and different hyperparams can impact it in different ways.
For example, EMC increases with epochs more rapidly for constant LR
than annealed LR, explaining our observations in Section 4.2.3.


# [Intuition 2] Temperature Scaling (TS) and Loss-Error Mismatch

Once a neural net has fit a training point,
the cross-entropy loss on it
can be lowered simply by upscaling the weights of the linear output layer.
This makes the model overconfident later in training,
as shown in [2].
For a noisy training point,
this leads to worse loss on similar test points.
The test error is not affected
as it depends only on the argmax of the class probabilities.
In high-noise settings,
test loss can worsen due to fitting noisy training points,
even as the test error improves from continued learning on clean points,
leading to loss-error mismatch.
TS fixes this by downscaling the logits.
Indeed, one finds that the temperature
(as obtained with a held-out set)
increases with epochs.
(see Fig. S1 in the Supplementary of [2]).


# [Intuition 3] Ens/SWA and Delayed Catastrophic Overfitting

We focus on test error as post-TS loss behaves similarly
and the intuitions transfer.
Flattening double descent is a special case
of delayed catastrophic overfitting,
as applied to the ascent to the peak.

From clean training points,
models learn generalizable patterns
and from noisy points, spurious ones which cause overfitting.
When noise is low,
the former dominates
and overfitting is benign.
Otherwise, overfitting is catastrophic.

The core intuition for Ens/SWA delaying catastrophic overfitting
is that generalizable patterns across checkpoints get reinforced,
while the spurious patterns cancel out.
Further intuition is that this is enabled by
the decision boundary being more "unstable" around noisy points
as compared to clean ones.
The experiments below substantiate these claims.


## Experiment 1
In Fig. 1, we see that
across epochs, the prediction flips
for a much higher fraction of the noisy points
than of the clean ones,
indicating higher instability.
The dataset here is CIFAR-10N Worst (~40% noise),
and training setup is same as in the paper.

## Experiment 2
Here, we measure the average predicted probability
for the clean and noisy subsets of CIFAR-10N Worst,
as proxy for the extent of memorization.
In Fig. 2 (a) and (b),
we find that SWA lowers the memorization
of clean points only a bit (\~0.1 probability),
but of noisy points by a lot (\~0.5 probability),
clearly establishing the differential effect.

## Experiment 3
Here, we replicate PHR on a synthetic dataset
with 2 input features,
with the aim of visualizing learnt decision surfaces
to solidify our intuitions.

We train 4-layer MLPs with 512 ReLU units per hidden layer on a 2-class spirals dataset of 1000 training points, with 20% of the labels flipped at random.
We train 16 MLPs and track the mean test error across epochs, as well as the test error of the ensemble (Fig. 3 (b)).

As per [3,4] Ens/SWA helps when the data has a "multi-view" structure,
or equivalently, the loss landscape has multiple modes.
This is hard to achieve for 2D dataset,
so instead we simulate the effect by training each MLP on a random 50% subsample
of the training data.

Fig. 3 (a) shows decision surfaces at epochs 440 and 1000 for 2 MLPs
and the ensemble.
Decision boundaries are spiky around noisy points
and smoother around clean ones.
While the generalizable parts of the spiral are retained in the ensemble,
the effects of noisy points are diminished.
Between epochs 440 and 1000,
individual models spike around noisy points more prominently
than they learn new parts of the spiral,
but the ensemble surface is relatively unchanged,
except for small improvements to learning the spiral.

We will further polish and incorporate the above in the paper.

# References

[1] Deep Double Descent: Where Bigger Models and More Data Hurt

[2] On Calibration of Modern Neural Networks

[3] Towards Understanding Ensemble, Knowledge Distillation and Self-Distillation in Deep Learning

[4] Deep Ensembles: A Loss Landscape Perspective

---

### Decision · Program_Chairs · 2024-09-25

**Decision:**

Accept (poster)

**Comment:**

**Summary**

This is an empirical study that shows that performance trends observed during training might reverse after post-hoc transforms like temperature scaling or stochastic weight averaging. This leads to the practical recommendation of *post-hoc selection*, i.e. to choose intermediate checkpoints based on their contribution to post-hoc performance rather than individual-run performance. Results are surprising, e.g. sometimes this method favors overfitting on individual runs.

**Recommendation**

This work is interesting for the research community and has some practical implications. I am happy to support its acceptance.